# Enhancing Adversarial Robustness on Categorical Data via Attribution Smoothing

## Abstract

Many efforts have been contributed to alleviate the adversarial risk of deep neural networks on continuous inputs. Adversarial robustness on general categorical inputs, especially tabular categorical attributes, has received much less attention. To echo this challenge, our work aims to enhance the robustness of classification over categorical attributes against adversarial perturbations. We establish an information-theoretic upper bound on the expected adversarial risk. Based on it, we propose an adversarially robust learning method, named Integrated Gradient-Smoothed Gradient *(IGSG)*-based regularization. It is designed to smooth the attributional sensitivity of each feature and the decision boundary of the classifier to achieve lower adversarial risk, i.e., desensitizing the categorical attributes in the classifier. We conduct an extensive empirical study over categorical datasets of various application domains. The experimental results confirm the effectiveness of *IGSG*, which surpasses the state-of-the-art robust training methods by a margin of approximately 0.4% to 12.2% on average in terms of adversarial accuracy, especially on high-dimension datasets.

## 1 Introduction

While categorical data widely exist in real-world safety-critical applications,much less research attention has been attracted to evasion attack and defense with categorical inputs, compared to the efforts with continuous data, e.g. images. It thus becomes a must to develop *adversarially robust learning paradigms* to harden ML systems with categorical inputs.Previous research on adversarially robust learning has mainly focused on enhancing the resilience of target classifiers against $L_Q$ and $L_\infty$ adversarial perturbations (Goodfellow et al., 2016; Madry et al., 2017; Moosavi-Dezfooli et al., 2019; Attias et al., 2019; Yin et al., 2019; Shafahi et al., 2019; Zhang et al., 2019; Wong et al., 2020; Bashivan et al., 2021; Zhang et al., 2022). However, when dealing with categorical data, the conventional Euclidean space framework used for continuous measurements, such as pixel intensities, is not a natural fit. Categorical variables like *race* and *occupation* have non-continuous and unordered qualitative values that cannot be combined in Cartesian products or ordered numerically.Thus, $L_0$-norm bounded adversarial perturbations are commonly employed to assess the robustness of categorical data (Lei et al., 2019; Bao et al., 2021).

Adversarial training (Madry et al., 2017) stands out as a predominant defense strategy in the continuous domain. However, adversarial training on categorical data poses a challenging Mixed Integer Nonlinear Programming (MINLP) problem (Lee & Leyffer, 2011). It involves the iterative generation of adversarial training samples within the categorical feature space, followed by model retraining using these adversarial samples in an alternating sequence. The exponential growth of the categorical adversarial space with increasing amounts of categorical features complicates the generation of adversarial samples via heuristic search like Brand-and-Bound (Pataki et al., 2010). In Section.3.1, we identify that exploring the categorical adversarial space leads to insufficient coverage, causing a distribution gap between adversarial training and future attacks, resulting in "robust overfitting" on categorical data (Rice et al., 2020). Encoding categorical features as one-hot vectors and relaxing the adversarial training to the continuous domain, treating one-hot vectors as probabilistic representations, partially mitigates categorical data complexities. However, this approach encounters a bottleneck— the non-convex and highly non-linear nature of the relaxed adversarial training objective, stemming from the bi-level mini-max training and deep neural network architectures. Consequently, the approximated solution lacks a bounded integrality gap to the original

discrete adversarial training problem, failing to guarantee optimality in the categorical feature space (Nohra et al., 2021). Thus, classifiers trained this way remain vulnerable to discrete adversarial samples in the combinatorial space. As empirically confirmed in Table.1, an MLP-based classifier tuned with the relaxed PGD-based adversarial training remains highly vulnerable to the state-of-the-art discrete adversarial attacks.

An alternative solution involves adversarial training within the embedding space of categorical variables. For instance, text classifiers can be defended using adversarial perturbations confined to the $L_Q$ ball around the target word in its embedding space (Zhu et al., 2019; Li et al., 2021; Pan et al., 2022). While effective for text-related tasks, this approach is unsuitable for general categorical data, such as system logs in cyber intrusion detection or medical examination records, lacking a meaningful embedding space. Additionally, domain-specific constraints crucial for adversarial perturbations, like synonymous words and semantic similarity measures, may be undefined or inapplicable across various categorical domains.

Considering the limitations of the discussed solutions, we seek an alternative strategy to mitigate the adversarial risk with categorical inputs. We focus on enforcing smoothness regularization on the target classifier (Ross & Doshi-Velez, 2018a; Finlay & Oberman, 2021). Specifically, our strategy first involves *penalizing the input gradients*. According to the information-theoretic upper bound on the expected adversarial risk on categorical data detailed in Section.3.2, penalizing input gradients mitigates

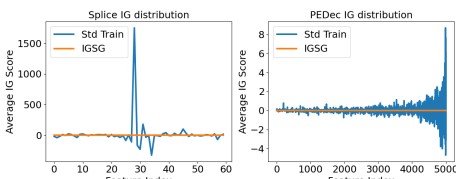

Figure 1: IG score distribution from the *IGSG* trained model and the undefended model *Std Train* on *Splice* and *PEDec* dataset.

the excessive curvature of the classification boundary and reduces the generalization gap of the target classifier. As a result, it alleviates the classifier's over-sensitivity to input perturbation. However, our comprehensive analysis indicates that merely penalizing the input gradient is not sufficiently secured. An additional influential factor is *the excessive reliance on specific features*, where a few features contribute significantly more to the decision output than others. The adversary may choose to perturb these dominant features to significantly mislead the classifier's output. To mitigate this, we propose to perform a Total-Variation (TV) regularization (Chambolle, 2004) on the integrated gradients (IG) of one-hot encoded categorical features. This evens the attribution from different features to the classification output. While IG is widely accepted as an XAI method to interpret feature-wise attribution to the classifier's decision output, our work is the first to uncover theoretically and empirically the link between smoothing the axiomatic attribution and improving adversarial robustness of the target classifier with categorical inputs. Combining both smoothing-driven regularization techniques, we propose Integrated Gradient-Smoothed Gradient (*IGSG*)-based regularization, effectively improving the adversarial robustness of the model. As shown in Figure.1, the IGSG-trained model demonstrates approximately evenly distributed IG scores for different categorical features. In contrast, the undefended model (*Std Train*) exhibits a highly skewed distribution of IG scores across features. Connecting Figure.1 with Figure.3, we observe that highly attacked features are precisely those with high IG scores. In summary, IGSG jointly smooths the classification boundary and desensitizes categorical features. It therefore prevents adversarial attacks from exploiting the over-sensitivity of the target classifier to the adversarial inputs.

Our technical contributions are summarized in the following perspectives:

**Understanding influencing factors of adversarial risk:** We've developed an information-theoretic upper bound to understand and minimize the expected adversarial risk on categorical data, providing insight into influential factors that can suppress adversarial risks effectively.

**Development of a model-agnostic robust training through regularized learning for categorical features.** We've reframed adversarial robustness, proposing IGSG, a method focused on minimizing our information-theoretic bound, enhancing feature contribution smoothness and decision boundary definitiveness during training. It's a universally adaptable solution for models dealing with categorical features.

**Extensive experimental study.** We've conducted thorough analyses comparing IGSG against the state-of-the-art adversarially robust training methods on three categorical datasets. The experimental results confirm the superior performances of models trained via IGSG.

## 2 RELATED WORKS

**Adversarial training** employs min-max optimization, generating adversarial samples via Fast Gradient Sign Method (FGSM) (Wong et al., 2020; Zhang et al., 2022) or Projected Gradient Descent (PGD) (Madry et al., 2017). TRADES (Zhang et al., 2019) optimizes a regularized surrogate loss, balancing accuracy and robustness. Adversarial Feature Desensitization (AFD) (Bashivan et al., 2021) leverages a GAN-like loss to learn invariant features against adversarial perturbations. While these methods can handle $L_1$-norm bounded adversaries for relaxed categorical data, ensuring consistent performance is uncertain. The challenge of "robust overfitting" in adversarial training (Rice et al., 2020) is addressed by Chen et al. (2020); Yu et al. (2022) in the continuous domain, but our investigation reveals this overfitting issue persists in the discrete feature space, unaddressed by existing continuous domain methods. Notably, our proposed IGSG successfully mitigates this problem.

**Adversarial learning for categorical data** typically involves search-based methods (Lei et al., 2019; Wang et al., 2020b; Bao et al., 2021; Li et al., 2018; Jin et al., 2020). However, the substantial time cost of generating adversarial samples hinders widespread application to general categorical data tasks, as seen in cybersecurity and medical services. Xu et al. (2023) suggested extending adversarial methods from continuous to discrete domains, but the MINLP nature of adversarial training poses challenges in generating sufficient samples for comprehensive defense. In text data, Ren et al. (2019) used word saliency and classification probability for guided word replacement, while methods like FreeLB Zhu et al. (2019); Li et al. (2021) applied multiple PGD steps to word embeddings. Dong et al. (2021) modeled the attack space as a convex hull of word vectors, and Wang et al. (2020a) enhanced BERT-based model robustness using information theory, often relying on language-specific constraints, limiting their broader applicability.

**Regularization-based methods** offer an alternative approach for enhancing adversarial robustness by penalizing the target classifier's complexity. Previous works Smilkov et al. (2017); Ross & Doshi-Velez (2018b); Finlay & Oberman (2021) proposed gradient magnitude regularization during training. Others Gu & Rigazio (2014); Jakubovitz & Giryes (2018); Hoffman et al. (2019) focused on penalizing the Frobenius norm of the Jacobian matrix for smoother classifier behavior. Additionally, Chen et al. (2019); Sarkar et al. (2021) suggested using Integrated Gradients (IG) for feature contribution measurement and applying regularization over IG to enhance robustness. Notably, these methods did not specifically target adversarial robustness. Our work reveals the effectiveness of IG-based regularization in adversarial robust training. Importantly, we demonstrate the significance of simultaneously regularizing gradient magnitude and IG distribution across different feature dimensions for a more potent approach.

## 3 UNDERSTANDING THE INFLUENCING FACTORS OF ADVERSARIAL RISK

**Preliminary.** Let's assume that a random sample $x_i = \{x_{i,1}, x_{i,2}, \ldots, x_{i,p}\}$ has $p$ categorical features and a class label $y_i$. Each feature $x_{i,j}$ can choose one out of $m$ possible category values. Following the one-hot encoding scheme, we can represent $x_i$ as a binary $\mathbb{R}^{p*m}$ matrix $b(x_i)$. Each row of $b(x_i)$ corresponds to the value chosen by feature $x_{i,j}$, i.e., $b(x_i)_{j,k^*} = 1$ when $x_{i,j}$ selects the $k^*$-th category value, and for all other $b(x_i)_{j,k \neq k^*} = 0$ ($k = 1, 2, ..., m$). An adversarial sample $\hat{x}_i = \{\hat{x}_{i,j,j=1,...,p}\}$ is generated by modifying the categorical values of a few features of $x_i$. The number of changed features from $x_i$ to $\hat{x}_i$ is noted as $diff(x_i, \hat{x}_i)$. Given a classifier $f$ and taking $b(x_i)$ as input to $f$, $f(b(x_i))$, simplified as $f(x_i)$, predicts its corresponding label $y_i$.

### 3.1 LIMITATIONS OF ADVERSARIAL TRAINING ON CATEGORICAL DATA

Firstly, we evaluate the limitations of adversarial training on categorical data. We implement $f$ as a Multilayer Perceptron (MLP) and conduct PGD-based adversarial training on it across three datasets. Subsequently, the resistance of $f$ to three evasion attacks is outlined in Table.1. With the attack budget 5 (i.e., $diff(x_i, \hat{x}_i) \leq 5$), both Forward Stepwise Greedy Search (FSGS) (Elenberg et al., 2018), and orthogonal matching pursuit based greedy search (OMPGS) (Wang et al., 2020b) can directly find attack samples $\hat{x}_i$. PGD attack in the 1-norm setting (PGD-1) (Madry et al., 2017) locates

Table 1: MLP with PGD-based adversarial training

| Dataset | Attack | Adv. Acc. | Defend |
|---------|--------|-----------|--------|
| Splice  | PGD-1  | 95.2%     | ✓      |
|         | OMPGS  | 51.7%     | ✗      |
|         | FSGS   | 43.6%     | ✗      |
| PEDec   | PGD-1  | 96.0%     | ✓      |
|         | OMPGS  | 74.1%     | ✗      |
|         | FSGS   | 52.5%     | ✗      |
| Census  | PGD-1  | 93.2%     | ✓      |
|         | OMPGS  | 62.7%     | ✗      |
|         | FSGS   | 54.1%     | ✗      |

attack samples and subsequently discretizes them to yield feasible adversarial samples $\hat{x}_i$. Table.1 show that the adversarially trained $f$ is only resilient against the PGD-1 based attack (high adversarial accuracy), remaining vulnerable facing the other two attacks (significantly lower adversarial accuracy). This suggests that the PGD-based adversarial training may not account for all possible adversarial samples, causing the model to overfit to the samples discovered by the PGD method.

Similar observations can be made for $f$ when using OMPGS-based adversarial training (see Figure.4 in Appendix.F). For the first 200 epochs, the adversarial accuracy and clean accuracy on the test set mirrored those on the training set. However, with further adversarial training, there is a notable increase in the adversarial accuracy and clean accuracy on the training set, while those on the test set remain unchanged, which indicates robust overfitting. The findings in Table.1 and Figure.4 show that the adversarial examples encountered during training do not generalize well to the test set. It suggests the presence of a distribution gap between discrete adversarial samples generated by different attack methods, as well as a distribution gap between adversarial samples generated during training and those encountered in the test set using the same attack method.

To provide further evidence of this distribution gap, we calculate the Wassernstein distance between the distributions of adversarial samples generated by PGD-1 and OMPGS on PGD/OMPGS-based adversarially trained model respectively (detailed in Appendix.F). A greater Wasserstein distance suggests a larger discrepancy between the two distributions. Two main observations are evident from Table.5. First, while PGD-based methods yield discrete adversarial samples with consistent distributions during both training and testing phases, these samples present significantly disparate distributions compared to those produced by OMPGS-based methods. This consistency in distribution with PGD-based methods is coherent with the results in Table.1, revealing substantial accuracy against PGD-based attacks but a lack of substantial defense against OMPGS-based attacks. Second, the adversarial samples derived via OMPGS exhibit a prominent distribution gap pre and post adversarial training. This distinction is indicative of the declining adversarial accuracy of the retrained classifier, as noted in Table.1 and Figure.4, through the course of the adversarial training.

**Robust overfitting with categorical vs. continuous data.** While robust overfitting in adversarial training with continuous data has been extensively researched Yu et al. (2022), the root causes differ when dealing with categorical data. Methods based on adversarial training typically employ heuristic search techniques like PGD or OMPGS to discover discrete adversarial samples for training. Due to the NP-hard nature of combinatorial search, these techniques can only explore a subset of adversarial samples, leaving samples outside this range to be perceived as Out-of-Distribution (OOD) by the classifier. This situation poses significant challenges for the model to generalize its robustness to unseen adversarial samples during testing. Attempted solutions such as thresholding out small-loss adversarial samples (Yu et al., 2022) have proven inadequate on categorical data in Appendix.I.4. Therefore, we opt for regularized learning-based paradigms for enhanced robustness in training with categorical data, avoiding the necessity to generate discrete adversarial samples.

## 3.2 INFORMATION-THEORETIC BOUND OF ADVERSARIAL RISK

Prior to developing our regularized learning approaches, we unveil the factors influencing adversarial risk for categorical data via the following analysis. We first define the adversarial risk.

**Definition 1.** *We consider a hypothesis space $\mathcal{H}$ and a non-negative loss function $\ell$: $\mu_z \times \mathcal{H} \to R^+$. Following (Xu & Raginsky, 2017; Asadi et al., 2018), given a training dataset $S^n$ composed of $n$ i.i.d training samples $z_i \sim \mu$, we assume a randomized learning paradigm $\mathcal{A}$ mapping $S^n$ to a hypothesis $f$, i.e., $f = \mathcal{A}(S^n)$, according to a conditional distribution $P_{f|S^n}$. The adversarial risk of $f$, noted as $\mathcal{R}_f^{adv}$, is given in Eq.1. It is defined as the expectation of the worst-case risk of $f$ on any data point $z = (x, y) \sim \mu_z$ under the $L_0$-based attack budget $diff(x, \hat{x}) \leq \epsilon$. The expectation is taken over the distribution of the $n$ training samples $S^n$ and the classifier $f = \mathcal{A}(S^n)$.*

$$\mathcal{R}_f^{adv} = \underset{S^n, P_{f|S^n}}{\mathbb{E}} \underset{z=(x,y)\sim\mu_z}{\mathbb{E}} \underset{diff(x,\hat{x})\leq\epsilon}{\sup} \ell(f(\hat{x}), y). \tag{1}$$

*As defined, $\mathcal{R}_f^{adv}$ measures the worst-case classification risk over an adversarial input $\hat{z} = (\hat{x}, y)$ where the attacker can modify at most $\epsilon$ categorical features. Similarly, we provide the empirical adversarial risk of $f$ in Eq.2. It is defined as the expectation of the worst-case risk over adversarial*

*samples $\hat{z} = (\hat{x}, y)$ over the joint distribution of $S^n$ and $P_{f|S^n}$.*

$$\hat{\mathcal{R}}_f^{adv} = \mathop{\mathbb{E}}_{S^n, P_{f|S^n}} \frac{1}{n} \sum_{z_i=(x_i,y_i)\in S^n} \sup_{diff(x_i,\hat{x}_i)\leq\epsilon} \ell(f(\hat{x}_i), y_i), \tag{2}$$

**Theorem 1.** *Let $\ell(f(x_i), y_i)$ be L-Lipschitz continuous for any $z_i = (x_i, y_i)$. Let $\mathcal{D}_f$ be the diameter of the hypothesis space $\mathcal{H}$. For each $x_i$, the categorical features modified by the worst-case adversarial attacker and the rest untouched features are noted as $\omega_i$ and $\overline{\omega_i}$, respectively. Given an attack budget $\epsilon$, the size of $\omega_i$ is upper bounded as $|\omega_i| \leq \epsilon$. The gap between the expected and empirical adversarial risk in Eq.1 and Eq.2 is bounded from above, as given in Eq.3.*

$$\mathcal{R}_f^{adv} - \hat{\mathcal{R}}_f^{adv} \leq \frac{L\mathcal{D}_f}{\sqrt{2}n} \sqrt{\sum_{i=1}^n I(f; z_i) + 2\sum_{i=1}^n \Psi(x_{i,\omega_i}, x_{i,\overline{\omega_i}}) + \sum_{i=1}^n \Phi(x_{i,\omega_i}, \hat{x}_{i,\omega_i})},$$

$$\Psi(x_{i,\omega_i}, x_{i,\overline{\omega_i}}) = |I(x_{i,\omega_i}; f) - I(x_{i,\overline{\omega_i}}, y_i; f)|, \tag{3}$$

$$\Phi(x_{i,\omega_i}, \hat{x}_{i,\omega_i}) = \alpha|I(\hat{x}_{i,\omega_i}; x_{i,\overline{\omega_i}}, y_i, f) - I(x_{i,\omega_i}; x_{i,\overline{\omega_i}}, y_i, f)|,$$

$$\alpha = \mathop{max}_{z_i=(x_i,y_i)\in S^n, |\omega_i|\leq\epsilon} 1 + \frac{|I(\hat{x}_{i,\omega_i}; x_{i,\overline{\omega_i}}, y_i) - I(x_{i,\omega_i}; x_{i,\overline{\omega_i}}, y_i)|}{|I(\hat{x}_{i,\omega_i}; x_{i,\overline{\omega_i}}, y_i, f) - I(x_{i,\omega_i}; x_{i,\overline{\omega_i}}, y_i, f)|},$$

*where $x_{i,\omega_i}$ and $\hat{x}_{i,\omega_i}$ are $\omega_i$ features before and after injecting adversarial modifications, and $I(X; Y)$ represents the mutual information between two random variables $X$ and $Y$.*

The proof can be found in Appendix.A. We further discuss the tightness of Eq.3 in Appendix.A. In the adversary-free case where $\hat{z} = z$, we show in Appendix.A that the bound established in Eq.3 is reduced to a tight characterization of generalization error for a broad range of models, which was previously unveiled in (Zhang et al., 2021; Bu et al., 2019).

The information-theoretical adversarial risk bound established in Eq.3 unveils two major factors to suppress the adversarial risk over categorical inputs.

**Factor 1. Reducing $I(f; z_i)$ for each training sample $z_i$ helps suppress the adversarial risk $f$.** $I(f, z_i)$ in Eq. 3 represents the mutual information between the classifier $f$ and each training sample $z_i$. Pioneering works (Xu & Raginsky, 2017; Bu et al., 2019; Zhang et al., 2021) have established that a lower value of $I(f, z_i)$ corresponds to a diminishing adversary-free generalization error. As widely acknowledged in adversarial learning research and emphasized in Eq. 3, a better generalizable classifier exhibits greater resilience to adversarial attacks, resulting in lower adversarial risk

**Factor 2. Reducing $\Psi(x_{i,\omega_i}, x_{i,\overline{\omega_i}})$ and $\Phi(x_{i,\omega_i}, \hat{x}_{i,\omega_i})$ helps smooth the feature-wise contribution to classification, thus reducing the adversarial risk.** We note that reducing the impact of excessively influential features can suppress adversarial risk, corresponding to minimizing the second and third terms beneath the square-root sign in Eq.3. **First**, in $\Psi(x_{i,\omega_i}, x_{i,\overline{\omega_i}})$, $I(x_{i,\omega_i}; f)$ and $I(x_{i,\overline{\omega_i}}, y_i; f)$ reflect the contribution of the feature subset $\omega_i$ and the rest features $\overline{\omega_i}$ to $f$. Features with higher mutual information have more substantial influence on the decision output, i.e. adversarially perturbing the values of these features is more likely to mislead the decision. Minimising $\Psi(x_{i,\omega_i}, x_{i,\overline{\omega_i}})$ thus decreases the contribution gap between the attacked and untouched features. It prompts the classifier to maintain a more balanced reliance on different features, thereby making it harder for adversaries to exploit influential features. **Second**, $\Phi(x_{i,\omega_i}, \hat{x}_{i,\omega_i})$ measures the sensitivity of features in $\omega_i$, in terms of how adversarial perturbations to this subset of features affect both the classification output and the correlation between $\omega_i$ and $\overline{\omega_i}$. Minimizing $\Phi(x_{i,\omega_i}, \hat{x}_{i,\omega_i})$ makes the classifier's output less sensitive to the perturbations over input features, which limits the negative impact of adversarial attacks. In conclusion, jointly minimising $\Psi(x_{i,\omega_i}, x_{i,\overline{\omega_i}})$ and $\Phi(x_{i,\omega_i}, \hat{x}_{i,\omega_i})$ ensures that the classifier does not overly rely on a few highly sensitive features. It helps reduce the susceptibility of the classifier to adversarial perturbation targeting at these features, which consequently limits the adversarial risk.Beyond the two factors, **minimizing the empirical adversarial risk $\hat{\mathcal{R}}_f^{adv}$ in Eq.3 may also reduce the adversarial risk.** This concept is synonymous with the principles of adversarial training. Nevertheless, as highlighted in Section.3.1, the efficacy of adversarial training is restricted.

## 4 *IGSG*: Robust Training for Categorical Data

Our design of adversarially robust training is in accordance with two recommended factors to minimize the adversarial risk. However, it is challenging to derive consistent estimates of mutual in-

formation between high-dimensional variables, e.g. model parameters of deep neural networks and high-dimension feature vectors, due to the curse of dimensionality Gao et al. (2018). Directly optimizing the mutual information-based bound is thus impractical. To overcome this bottleneck, we propose the *IGSG*-based robust training paradigm. It jointly applies two smoothness-enhancing regularization techniques into the learning process of a classifier with categorical inputs, in order to mitigate the adversarial attack over categorical data.

**Minimizing $I(f; z_i)$ by smoothing the curvature of the classification boundary.** In previous work, Fisher information $\rho(z_i)_f$ was utilized as a quantitative measure of the information that the hypothesis $f$ contains about the training sample $z_i$ (Hannun et al., 2021). As shown in Wei & Stocker (2016), $\rho(z_i)_f$ is closely related to the mutual information $I(f; z_i)$, higher/lower $\rho(z_i)_f$ indicates higher/lower $I(f; z_i)$. Our work aims to minimize $\rho(z_i)_f$ to effectively penalize excessively high mutual information $I(f; z_i)$. The computation of $\rho(z_i)_f$ is detailed in Eq.16 of (Hannun et al., 2021). In this context, suppressing $\rho(z_i)_f$ (approximately suppressing $I(f; z_i)$) is equivalent to penalizing the magnitude of the gradient of the loss function with respect to each $z_i$. This approach, supported by findings in (Smilkov et al., 2017), uses gradient regularization to smooth the classifier's decision boundary, thereby reducing the potential risk of overfitting and enhancing adversarial resilience We calculate the gradient of the classification loss to the one-hot encoded representation of $b(x_i)$, which gives as $\nabla_{b(x_i)}\ell(x_i, y_i; \theta) \in \mathbb{R}^{p*m}$. Each element of $\nabla_{b(x_i)}\ell(x_i, y_i; \theta)$ is formulated as $\frac{\partial}{\partial b(x_i)_{j,k}}\ell(x_i, y_i; \theta)$. According to (Yang et al., 2021), $\nabla_{b(x_i)}\ell(x_i, y_i; \theta)$ measures the curvature of the decision boundary around the input. A larger magnitude of $\nabla_{b(x_i)}\ell(x_i, y_i; \theta)$ indicates a more twisted decision boundary, thus a less stable decision around the input. Enforcing the regularization over the magnitude $\|\nabla_{b(x_i)}\ell(x_i, y_i; \theta)\|_q$ leads to a smoother decision boundary (with lower curvature) and improves the robustness of the decision output $f(x_i)$ against potential perturbation. In this work, we apply smoothed Gradient Regularization (SG) (Smilkov et al., 2017) to further boost the smoothness of the classifier.

**Minimizing $\Psi(x_{i,\omega_i}, x_{i,\overline{\omega_i}})$ and $\Phi(x_{i,\omega_i}, \hat{x}_{i,\omega_i})$ via smoothing the distribution of feature-wise contribution to the classification output.** Minimizing these terms involves evaluating the mutual information between the feature subset $\omega_i$ and the combined set of remaining features and the trained model $f$. Approximating this mutual information-based penalization with Fisher information is thus infeasible. The primary goal of regularizing these terms is to prevent the classifier from relying too heavily on a few influential features. To achieve this, we propose using Integrated Gradient (IG) (Sundararajan et al., 2017) to assess feature-wise contributions to the classification output. We apply Total-Variance (TV) regularization over the feature-wise Integrated Gradient to promote a smooth and balanced distribution of feature-wise attribution. In Appendix.I.1, we show empirically with toy models that performing the proposed TV regularization can reduce the estimated value of both mutual information-based terms.

We extend the computation of the IG scores in the categorical feature space by first defining a baseline input $x'$. We augment the set of optional category values for each feature $x_{i,j}$: *we add one dummy category* $m+1$, with constantly all 0 values for the embedding vector in $f$. Each feature of $x'$ is set to take the dummy category value, i.e., $b(x')_{j,m+1} = 1, b(x')_{j,k} = 0(k = 1, 2, \ldots, m)$. By feeding $b(x')$ to the classifier, no useful information is conveyed for classification, making it a non-informative baseline. Given the defined baseline input $x'$, the IG score of each categorical feature $x_{i,j}$ is approximated as:

$$IG(x_i)_j = \sum_{k=1}^{m} IG(x_i)_{j,k} = \sum_{k=1}^{m} (b(x_i)_{j,k} - b(x')_{j,k}) \times \frac{1}{T}\sum_{t=1}^{T} \frac{\partial f(b(x') + \frac{t}{T} \times [b(x_i) - b(x')])}{\partial b(x_i)_{j,k}} \quad (4)$$

where $T$ is the number of steps in the Riemman approximation of the integral. We empirically choose $T$=20, which provides consistently good learning performances. $IG(x_i)_j$ derived along the trajectory between $b(x')$ and $b(x_i)$ hence represents the contribution of $x_{i,j}$ to the classifier's output.

To ensure a smooth and balanced distribution of IG scores and to mitigate excessive dependency on specific features, we propose to minimize the TV loss of the normalized IG scores, as influenced by prior work (Chambolle, 2004). Initially, we employ a softmax transformation to normalize the IG scores of each feature $x_{i,j}$, ensuring the normalized scores lie within $[0, 1]$ and collectively sum to 1. The TV regularization term is then defined as the sum of the absolute differences between neighboring features' normalized IG scores: $\ell_{TV}IG(x_i) = \sum_{j=1}^{p-1}|IG(x_i)_j - IG(x_i)_{j+1}|$, following the

TV loss used in time series data analysis (Chambolle, 2004). This minimization promotes a more balanced distribution of feature-wise contributions to the classifier's decision.

Combining Eq.18 in Appendix.E and $\ell_{TV}IG(x_i)$, the objective function of IGSG gives:

$$\min_{\theta} \mathbb{E}_{(x_i,y_i)\in S^n} \ell(x_i,y_i;\theta) + \alpha\ell_{TV}IG(x_i) + \frac{\beta}{R}\sum_{r=1}^{R}||G_r||_p$$
$$\text{where } G_{r,j,k} = \frac{\partial}{\partial b(x_r)_{j,k}}\ell(x_r,y_i;\theta) - \frac{\partial}{\partial b(x_r)_{j,k^*}}\ell(x_r,y_i;\theta)$$

(5)

where $\alpha$ and $\beta$ are hyper-parameters set by cross-validation.

## 5 EXPERIMENTAL EVALUATION

### 5.1 EXPERIMENTAL SETUP

**Summary of datasets.** To evaluate the proposed IGSG algorithm, we employ two categorical datasets and one mixed dataset with both categorical and numerical features, each from different applications and varying in the number of samples and features.
1) Splice-junction Gene Sequences (Splice) (Noordewier et al., 1990). The dataset includes 3190 gene sequences, each with 60 categorical features from the set {A, G, C, T, N}. Each sequence is labeled as intron/exon borders (*IE*), exon/intron borders (*EI*), or neither.
2) Windows PE Malware Detection (PEDec) (Bao et al., 2021). This dataset, used for PE malware detection, consists of 21,790 Windows executable samples, each represented by 5,000 binary features denoting the presence or absence of corresponding malware signatures. The samples are categorized as either benign or malicious.
3) Census-Income (KDD) Data (Census) (Lane & Kohavi, 2000). This dataset includes census data from surveys conducted from 1994 to 1995, encompassing 299,285 samples. Each has 41 features related to demographics and employment, with 32 categorical and 9 numerical. The task is to determine whether subjects fall into the low-income (less than \$50,000) or high-income group.

For *Splice* and *PEDec*, we use 90% and 10% of the data samples as the training and testing set to measure the adversarial classification accuracy. For *Census*, we use the testing and the training set given by (Lane & Kohavi, 2000), i.e., 199,523 for training and 99,762 for testing.

**Robustness evaluation protocol.** Three domain-agnostic attack methods, FSGS (Elenberg et al., 2018), OMPGS (Wang et al., 2020b) and PCAA (Xu et al., 2023), designed specifically for generating discrete adversarial perturbations in categorical data, are employed to evaluate adversarial robustness. Due to the discontinuous nature of categorical data, traditional attacks like PGD and FGSM cannot be directly applied. Further discussion is presented in Appendix.G. FSGS, OMPGS and PCAA, with proven attack effectiveness across various real-world applications, are suitable for comparing the effectiveness of different robust model training methods on categorical input.

We traverse varied attack budgets (the maximum number of the modified features) for OMPGS attacks. Due to the high computational complexity of FSGS (Bao et al., 2021), we set a fixed attack budget of 5 on all three datasets. For PCAA, we also fix the attack budget to be 5. On each dataset, we use MLP and Transformer (Vaswani et al., 2017) as the target classifier. Due to space limitations, we provide detailed attack settings in Appendix.H.1, the experimental results on Transformer models in Appendix.I.3, and the experimental results of PCAA attack in Appendix.I.7

**Baselines.** We involve one undefended model and 7 state-of-the-art robust training methods as the baselines in the comparison with *IGSG*. Specifically, we include 5 adversarial training baselines Adv Train (Madry et al., 2017), Fast-BAT (Zhang et al., 2022), TRADES (Zhang et al., 2019), AFD (Bashivan et al., 2021) and PAdvT (Xu et al., 2023), and 2 regularization-based baselines IGR (Ross & Doshi-Velez, 2018b) and JR (Hoffman et al., 2019). The details of the baselines can be found in Appendix.H.2 and the details of the hyper parameter settings can be found in Appendix.H.3.

**Performance metrics.** We compare the *adversarial accuracy* of the target models trained using the methods above against FSGS and OMPGS attacks. We evaluate the adversarial robustness of mixed-type datasets by attacking categorical features with FSGS/OMPGS and numerical features with PGD-$\infty$. Further details can be found in Appendix.H.4. Time complexity analysis and training time

Table 2: Adversarial Accuracy under FSGS attack and Accuracy (%) for *IGSG* and baseline models. Adv Train (Madry et al., 2017), Fast-BAT (Zhang et al., 2022), TRADES (Zhang et al., 2019), AFD (Bashivan et al., 2021), PAdvT (Xu et al., 2023), IGR (Ross & Doshi-Velez, 2018b), JR (Hoffman et al., 2019)

| Dataset | Attack | Undefended | Adversarial Training baselines | | | | | Regularization baselines | | Ours |
|---|---|---|---|---|---|---|---|---|---|---|
| | | Std Train | Adv Train | Fast-BAT | TRADES | AFD | PAdvT | IGR | JR | IGSG |
| Splice | budget=5 | 36.7±4.8 | 43.6±0.7 | 28.7±7.4 | 23.3±8.6 | 21.1±13.0 | 39.1±1.7 | 40.9±3.0 | 4.3±3.7 | **44.0±2.6** |
| | Clean | 95.2±2.5 | 96.2±0.4 | 95.6±1.0 | 96.3±0.3 | 93.4±0.7 | 94.9±1.3 | 95.2±0.6 | 95.2±0.9 | 95.9±0.7 |
| PEDec | budget=5 | 14.9±0.8 | 53.1±1.7 | 62.4±2.7 | 31.0±2.5 | 74.3±3.9 | 46.9±2.9 | 31.4±0.9 | 74.3±0.2 | **86.5±3.8** |
| | Clean | 96.4±0.2 | 96.2±0.0 | 96.2±0.1 | 96.4±0.1 | 96.0±0.2 | 96.5±0.3 | 96.4±0.0 | 95.4±0.1 | 95.5±0.2 |
| Census | budget=5 | 46.2±1.8 | 54.1±2.3 | 63.4±3.8 | 49.8±1.6 | 60.2±1.9 | 61.9±5.4 | 45.8±1.7 | 48.3±3.4 | **67.2±3.5** |
| | Clean | 95.4±0.1 | 94.5±0.3 | 95.0±0.1 | 94.8±0.3 | 95.2±0.2 | 95.2±0.1 | 95.3±0.1 | 95.4±0.1 | 95.5±0.2 |

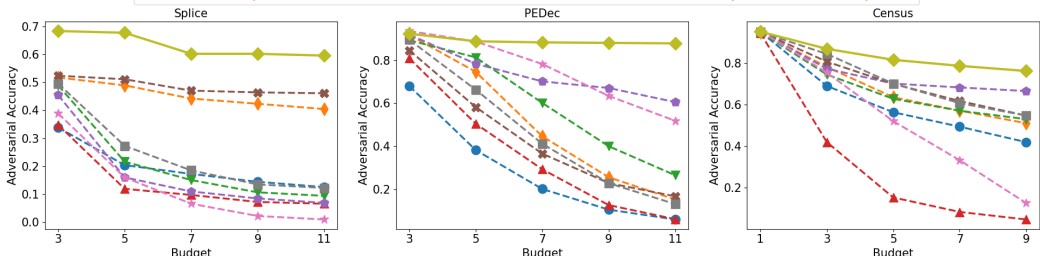

Figure 2: Adversarial accuracy for *IGSG* and baselines under OMPGS attack with varied budgets.

for different methods are provided in Appendix.I.5. The code is available at https://github.com/fshafrh/IGSG.

## 5.2 EXPERIMENTAL RESULTS

**Adversarial Accuracy Performance of *IGSG* Compared to Baseline Methods.** Table.2 reports the accuracy and the adversarial accuracy against FSGS attacks for each robust training method. From the results, we can see that the adversarial accuracy of *IGSG* significantly outperforms the baseline methods. Especially, on *PEDec*, *IGSG* can largely improve the adversarial accuracy up to 86.5%. In comparison, the best baseline of robust training, *JR* and *AFD*, only achieves an adversarial accuracy score of 74.3%. *IGSG* also achieves comparable accuracy on the three datasets.

Figure 2 illustrates the adversarial accuracy of all the methods tested under OMPGS-based attacks with varying attack budgets. Higher attack budgets indicate stronger attacks against the targeted classifier, resulting in lower adversarial accuracy overall. Similar to the undefended model, most baseline methods experience a decline in adversarial accuracy as the attack strength increases. In contrast, the proposed method, IGSG, consistently achieves higher and more stable levels of adversarial accuracy across all three datasets. Specifically, on *PEDec*, IGSG maintains an adversarial accuracy above 88% regardless of the attack strength. On *Splice*, IGSG consistently outperforms other baseline methods, exhibiting a performance gain of over 10%. On *Census*, IGSG initially shows similar adversarial accuracy to other baselines under small attack budgets but demonstrates a significantly slower rate of decline as the attack budget increases. Notably, adversarial training methods like *Adv Train* perform poorly on *PEDec*. This is because the feature space of *PEDec* is extensive, causing adversarial training to suffer from robust overfitting on categorical data. The attack can only explore a small fraction of all possible adversarial perturbations, limiting the effectiveness of adversarial training, while *IGSG* can provide consistently robust classification regardless of the feature dimensionality. *JR* performs well on *PEDec*, while the performance on *Splice* and *Census* is constantly bad. Using regularization as well, *IGSG* has a more stable performance on different datasets. It is worth noting that *Splice* has a few particularly sensitive features. Modifying these features can result in a change in whether a sample crosses an intron/exon or exon/intron boundary, or neither physically, which causes misclassification. Thus, all the defense methods involved in the test do not perform well against attacks on *Splice*.

**Ablation Study.** We include the following variants of the proposed *IGSG* method in the ablation study. ***SG*** and ***IG*** are designed to preserve only the smoothed gradient-based (*SG*, see Eq.18) or the IG-based smoothness regularization (*IG*, see Eq.17) respectively in the learning objective. We compare *SG* and *IG* to *IGSG* for demonstrating the advantage of simultaneously performing the IG and gradient smoothing-based regularization. ***IGSG-VG***: We replace the smoothed gradient given

in Eq.18 with the vanilla gradient of the one hot tensor. Another four variants to provide additional validation for the design of *IGSG* are presented in Appendix.I.6

Table.3 shows that *IGSG* consistently outperforms the variants in adversarial accuracy against both FSGS and OMPGS attacks, affirming the effectiveness of *IGSG*'s design in mitigating both types of greedy search-based attacks simultaneously. *SG* does not employ IG-based regularization, resulting in a classifier that may overly rely on a few highly influential features contributing most to the classification out-

Table 3: Ablation Study. Adversarial Accuracy and Accuracy (%) for *IGSG* variants with an attack budget of 5.

| Dataset | Adversary | SG | IG | IGSG-VG | IGSG |
|---|---|---|---|---|---|
| Splice | FSGS | 43.3±3.0 | 40.3±5.0 | 39.7±2.4 | **44.0±2.6** |
| | OMPGS | 59.9±6.5 | 54.9±4.9 | 59.4±5.3 | **63.8±4.2** |
| | Clean | 95.7±0.5 | 94.7±1.0 | 95.2±1.1 | 95.9±0.7 |
| PEDec | FSGS | 12.7±1.8 | 84.2±2.9 | 81.6±3.8 | **86.5±3.8** |
| | OMPGS | 28.6±1.1 | 83.4±7.6 | 82.3±3.5 | **88.0±4.0** |
| | Clean | 96.4±0.1 | 94.8±0.3 | 95.2±0.2 | 95.5±0.2 |
| Census | FSGS | 47.9±2.1 | 57.8±0.8 | 54.1±1.6 | **67.2±3.5** |
| | OMPGS | **71.4±7.8** | 65.9±2.7 | 69.3±6.4 | 71.3±9.0 |
| | Clean | 95.1±0.3 | 95.5±0.1 | 95.4±0.0 | 95.5±0.2 |

put. These sensitive features can be readily targeted by both types of greedy search-based attacks, particularly on *PEDec*. In comparison, *IG* lacks the classification boundary smoothness, leading to a slight decrease in performance compared to *IGSG*. The results with *SG* and *IG* show that the two attributional smoothness regularization terms employed by *IGSG* are complementary to each other in improving the adversarial robustness of the built model.

*IGSG-VG* replaces the smoothed gradient-based regularization defined in Eq.18 and Eq.19 with a vanilla gradient. Its diminished performance shows the merit of introducing the smoothed gradient computing and the mean field smoothing based technique in Eq.18 and Eq.19.

Table 4: MLP with IGSG training and Performance Gain Compared to PGD-based Adversarial Training

| Dataset | Attack | Adv. Acc. | Gain |
|---|---|---|---|
| Splice | PGD-1 | 95.6% | 0.4% ∼ |
| | OMPGS | 63.8% | 12.1% ↑ |
| | FSGS | 44.0% | 0.4% ∼ |
| PEDec | PGD-1 | 94.5% | -1.5% ∼ |
| | OMPGS | 88.0% | 13.9% ↑ |
| | FSGS | 86.5% | 34% ↑ |
| Census | PGD-1 | 93.0% | -0.2% ∼ |
| | OMPGS | 71.3% | 8.6% ↑ |
| | FSGS | 67.2% | 13.1% ↑ |

**Effectiveness of Avoiding Robust Overfitting.** By utilizing regularization, IGSG avoids the issue of "robust overfitting" encountered in adversarial training. This results in improved performance, as demonstrated in Table.4, compared to the adversarial accuracy shown in Table.1. We conduct the comparison between IGSG and two works mitigating robust overfitting in continuous domain (Chen et al., 2020; Yu et al., 2022). IGSG achieves consistently better adversarial robustness. The details are presented in Appendix.I.4

**Reduced Attack Frequency with IGSG.** We compare the frequency of each feature attacked under OMPGS on *Splice* and *PEDec*. The attack frequency represents the number of times a feature appears among the altered features in all successful adversarial attack samples. As seen in Figure.3, *IGSG* results in fewer and lower peaks on *Splice* compared to the undefended model, in-

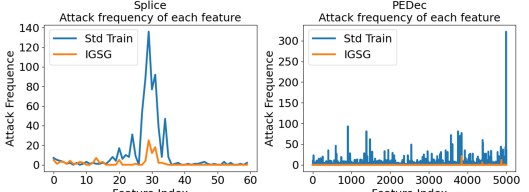

Figure 3: Attack frequency reduced by IGSG

dicating enhanced robustness. For *PEDec*, the feature with the highest attack frequency is entirely suppressed with *IGSG*. This demonstrates the effectiveness of *IGSG*, with feature desensitization being achieved post-training.

## 6 CONCLUSION

In this work, we first unveil influencing factors of adversarial threats on categorical inputs via developing an information-theoretic upper bound of the adversarial risk. Guided by the theoretical analysis, we further propose *IGSG*-based adversarially robust model training via enforcing the two smoothness regularization techniques on categorical data, which helps mitigate adversarial attacks on categorical data. On the one hand, our method smooths the influence of different categorical features and makes different features contribute evenly to the classifier's output. On the other hand, our method smooths the decision boundary around an input discrete instance by penalizing the gradient magnitude. We demonstrate the domain-agnostic use of *IGSG* across different real-world applications. In our future study, we will extend the proposed method to the text classification task and compare it with text-specific robust training methods enhanced with semantic similarity knowledge.

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
