# A  PROOF TO THEOREM.1

**Definition 2.** *Diameter of $f$: Assuming that the hypothesis space $\mathcal{H}$ is a bounded banach space, the diameter of $f \in \mathcal{H}$ is defined as:*

$$\mathcal{D}_f = \sup_{f,f' \in \mathcal{H}} d(f, f') \tag{6}$$

*where $d$ is the distance metric of $\mathcal{H}$.*

**Definition 3.** *Lipschitz continuousity of $\ell$: Assuming that $\ell(f(x_i), y_i)$ is L-Lipschitz for any $z_i = (x_i, y_i)$, the following inequality holds for any $f$ and $f'$ in $\mathcal{H}$:*

$$|\ell(f(x_i), y_i) - \ell(f'(x_i), y_i))| \le L\, d(f, f') \tag{7}$$

**Proof to Eq.3:** Given $\mu_z$ and a classifier $f$ trained using $S^n$, we assume the distribution of the worst-case adversarial samples of $f$ as $\hat{\mu}_z$, determined by $\mu_z$ and $f$ jointly. Any worst-case adversarial sample $\hat{z}_i$ derived by solving the loss maximization problem $\arg \max_{\text{diff}(\hat{z}_i, z_i) \le \epsilon} \ell(f(x_i), y_i)$ can be thus considered as a sample from $\hat{\mu}_{\hat{z}}$. We can then extend the Total Variation (TV) distance-based generalization bound of $f$, which is established by Theorem.2 in (Zhang et al., 2021) as below:

$$\mathrm{E}_f[\mathcal{R}_f^{adv}] \le \mathrm{E}_f[\hat{\mathcal{R}}_f^{adv}] + L\,\mathcal{D}_f\, \mathbb{TV}(P_f \times \hat{\mu}_{\hat{z}}, P_{f \times \hat{z}_i}) \tag{8}$$

where $\mathbb{TV}(\cdot, \cdot)$ denotes the Total Variation distance between two probabilistic distribution. $P_f$ and $\hat{\mu}_{\hat{z}}$ are the marginal distribution of $f$ and the worst-case adversarial sample $\hat{z}_i$. $P_{f \times \hat{z}_i}$ denotes the joint distribution of $f$ and $\hat{z}_i$.

Pinsker's inequality in information theory (Cover & Thomas, 2005) gives further the upper bound of the Total-Variation distance: $\mathbb{TV}(P_f \times \hat{\mu}_{\hat{z}}, P_{f \times \hat{z}_i}) \le \sqrt{\frac{D_{KL}(P_{f,\hat{z}_i}, P_f \times P_{\hat{z}_i})}{2}} = \sqrt{\frac{I(f, \hat{z}_i)}{2}}$, where $D_{KL}$ is the KL divergence between the two probabilistic distributions. Based on this, we can further formulate Eq.8 by letting $z = z_i$ ($i$=1,2,3,...,n) and using mutual information between $f$ and $\hat{z}_i$:

$$\begin{aligned}
\mathrm{E}_f[\mathcal{R}_f^{adv}] &\le \mathrm{E}_f[\hat{\mathcal{R}}_f^{adv}] + \frac{L\,\mathcal{D}_f}{\sqrt{2}n} \sqrt{\sum_{i=1}^{n} I(f; \hat{z}_i)} \\
&\le \mathrm{E}_f[\hat{\mathcal{R}}_f^{adv}] + \frac{L\,\mathcal{D}_f}{\sqrt{2}n} \sqrt{\sum_{i=1}^{n} I(f; z_i) + \sum_{i=1}^{n}(I(f; \hat{z}_i) - I(f; z_i))}
\end{aligned} \tag{9}$$

where $\{z_i = (x_i, y_i)\} \in S^n$ are statistically independent training samples and $\hat{z}_i$ the corresponding worst-case adversarial sample. We can extend $I(f; \hat{z}_i) - I(f; z_i)$ as below. In this study, we only consider feature perturbation and exclude label flipping attacks from the proposed attack scenario. We first split $\hat{z}_i = (\hat{x}_i, y_i)$ and $z_i = (x_i, y_i)$ into $\hat{z}_i = (\hat{x}_{i,\omega_i}, x_{i,\overline{\omega}_i}, y_i)$ and $\hat{z}_i = (\hat{x}_{i,\omega_i}, x_{i,\overline{\omega}_i}, y_i)$ respectively. Since features in $\overline{\omega}_i$ remain untouched in the attack, we use the same notation of these unmodified features in $\hat{z}_i$ and $z_i$.

$$\begin{aligned}
& I(f; \hat{z}_i) - I(f; z_i) \\
=\ & I(f; \hat{x}_{i,\omega_i}, x_{i,\overline{\omega}_i}, y_i) - I(f; x_{i,\omega_i}, x_{i,\overline{\omega}_i}, y_i) \\
=\ & I(x_{i,\overline{\omega}_i}, y_i; f) - I(x_{i,\omega_i}; f) + I(\hat{x}_{i,\omega_i}; f | x_{i,\overline{\omega}_i}, y_i) - I(x_{i,\overline{\omega}_i}, y_i; f | x_{i,\omega_i}) \\
=\ & I(x_{i,\overline{\omega}_i}, y_i; f) - I(x_{i,\omega_i}; f) + H(\hat{x}_{i,\omega_i} | x_{i,\overline{\omega}_i}, y_i) + H(f | x_{i,\overline{\omega}_i}, y_i) - H(\hat{x}_{i,\omega_i}, f | x_{i,\overline{\omega}_i}, y_i) \\
& - H(x_{i,\overline{\omega}_i}, y_i | x_{i,\omega_i}) - H(f | x_{i,\omega_i}) + H(x_{i,\overline{\omega}_i}, y_i, f | x_{i,\omega_i}) \\
=\ & I(x_{i,\overline{\omega}_i}, y_i; f) - I(x_{i,\omega_i}; f) + H(\hat{x}_{i,\omega_i}) - I(\hat{x}_{i,\omega_i}; x_{i,\overline{\omega}_i}, y_i) + H(f) - I(x_{i,\overline{\omega}_i}, y_i; f) \\
& - H(x_{i,\overline{\omega}_i}, y_i) + I(x_{i,\omega_i}; x_{i,\overline{\omega}_i}, y_i) - H(f) + I(x_{i,\omega_i}; f) \\
& - H(\hat{x}_{i,\omega_i}, f | x_{i,\overline{\omega}_i}, y_i) + H(x_{i,\overline{\omega}_i}, y_i, f | x_{i,\omega_i}) \\
=\ & I(x_{i,\overline{\omega}_i}, y_i; f) - I(x_{i,\omega_i}; f) + H(\hat{x}_{i,\omega_i}) - I(\hat{x}_{i,\omega_i}; x_{i,\overline{\omega}_i}, y_i) + H(f) - I(x_{i,\overline{\omega}_i}, y_i; f) \\
& - H(x_{i,\overline{\omega}_i}, y_i) + I(x_{i,\omega_i}; x_{i,\overline{\omega}_i}, y_i) - H(f) + I(x_{i,\omega_i}; f) \\
& - H(f | x_{i,\overline{\omega}_i}) - H(\hat{x}_{i,\omega_i} | x_{i,\overline{\omega}_i}, f) + H(x_{i,\overline{\omega}_i}, f | x_{i,\omega_i}) \\
\le\ & 2|I(x_{i,\omega_i}; f) - I(x_{i,\overline{\omega}_i}, y_i; f)| + |I(\hat{x}_{i,\omega_i}; x_{i,\overline{\omega}_i}, y_i, f) \\
& - I(x_{i,\omega_i}; x_{i,\overline{\omega}_i}, y_i, f)| + |I(\hat{x}_{i,\omega_i}; x_{i,\overline{\omega}_i}, y_i) - I(x_{i,\omega_i}; x_{i,\overline{\omega}_i}, y_i)|
\end{aligned}$$

$$\tag{10}$$

where $H(X|Y)$ and $I(X;Y|Z)$ denotes the conditional entropy of a random variable $X$ given the other random variable $Y$ and the conditional mutual information between $X$ and $Y$ given another random variable $Z$. By introducing $\alpha = \max\limits_{z_i=(x_i,y_i)\in S^n} 1 + \frac{|I(\hat{x}_{i,\omega_i};x_{i,\overline{\omega_i}},y_i)-I(x_{i,\omega_i};x_{i,\overline{\omega_i}},y_i)|}{|I(\hat{x}_{i,\omega_i};x_{i,\overline{\omega_i}},y_i,f)-I(x_{i,\omega_i};x_{i,\overline{\omega_i}},y_i,f)|}$ to Eq.10, we can derive Eq.3.

**We discuss about the tightness of the bound in Eq.3 from the following perspectives**. First, we show this bound reduces to a individual sample based upper bound of the generalization error of $f$ in the adversary-free case. It converges to zero when $n \to \infty$ with the same speed as that established in Proposition.1 of Bu et al. (2019). This bound enjoys a close level of tightness *in the adversary-free scenario* as that proposed in in Bu et al. (2019).

We first give the definition of the expected and empirical risk under the adversary-free setting, following Definition.1.

**Definition 4.** *Following (Xu & Raginsky, 2017; Asadi et al., 2018), given a training dataset $S^n$ composed of $n$ i.i.d training samples $z_i \sim \mu$, we assume a randomized learning paradigm $\mathcal{A}$ mapping $S^n$ to a hypothesis $f$, i.e., $f = \mathcal{A}(S^n)$, according to a conditional distribution $P_{f|S^n}$. The expected classification risk of $f$ under the adversary-free scenario, noted as $\mathcal{R}_f$, gives in Eq.11. The expectation is taken over the distribution of the n training samples $S^n$ and the classifier $f = \mathcal{A}(S^n)$.*

$$\mathcal{R}_f = \mathop{\mathbb{E}}\limits_{S^n, P_{f|S^n}} \mathop{\mathbb{E}}\limits_{z=(x,y)\sim\mu_z,} \ell(f(x), y). \tag{11}$$

*Similarly, we provide the empirical risk of $f$ under the adversary-free scenario in Eq.12. It is taken as the expectation over the distribution of the n training samples and the classifier.*

$$\hat{\mathcal{R}}_f = \mathop{\mathbb{E}}\limits_{S^n, P_{f|S^n}} \frac{1}{n} \sum_{z_i=(x_i,y_i)\in S^n} \ell(f(x_i), y_i) \tag{12}$$

With the adversary-free setting, $\hat{x} = x$. This makes $\Phi(x_{i,\omega_i}, \hat{x}_{i,\omega_i})$ vanish as $I(\hat{x}_{i,\omega_i}; x_{i,\overline{\omega_i}}, y_i, f) = I(x_{i,\omega_i}; x_{i,\overline{\omega_i}}, y_i, f)$. Similarly, $\Psi(x_{i,\omega_i}, x_{i,\overline{\omega_i}}) = |I(x_{i,\omega_i}; f) - I(x_{i,\overline{\omega_i}}, y_i; f)|$ is reduced to $I(z_i; f)$, since $\omega_i = \emptyset$ for each training sample $z_i$. As a result, the bound given in Eq.3 shrinks to the following form in Eq.13:

$$\mathcal{R}_f - \hat{\mathcal{R}}_f \leq \frac{\sqrt{3}\,L\,\mathcal{D}_f}{\sqrt{2}n} \sqrt{\sum_{i=1}^{n} I(f; z_i)}. \tag{13}$$

where $\mathcal{R}_f$ and $\hat{\mathcal{R}}_f$ are expected and empirical risk under the adversary-free setting. In comparison, Proposition.1 (Eq.19 and 20) in Bu et al. (2019) provides the upper bound of the generalization error of $f$ in a similar form:

$$\mathcal{R}_f - \hat{\mathcal{R}}_f \leq \frac{1}{n} \sum_{i=1}^{n} \sqrt{2R^2 I(f; z_i)}. \tag{14}$$

with the condition that the loss function $\ell(f, z)$ is $R$-sub-Gaussian under $z \sim \mu_z$ for all $f \in \mathcal{H}$. We can find that the two adversary-free bounds in Eq.13 and Eq.14 only differ in the scaling constant. When $n$ (the number of training samples) goes to infinity, both bounds vanish with the same convergence speed. Compared to the training set mutual information $I(f; S^n)$ based bound proposed Theorem.1 of Xu & Raginsky (2017), the individual sample mutual information-based bound (Eq.13 and Eq.14) poses a tighter bound over the generalization error according to the theoretical and empirical analysis conducted in Bu et al. (2019). In Xu & Raginsky (2017), the information-theoretic bound is built by assuming that the loss function $\ell(f, z)$ has a bounded cumulative generating function with $z \sim \mu_z$ and $f \in \mathcal{H}$. Nevertheless, this assumption does not necessarily hold. Our study thus avoids this shortcoming and adopts the individual sample mutual information to develop the adversarial risk analysis. In conclusion, we develop theoretical analysis under a more general condition about the cumulative generating function of the loss function compared to Xu & Raginsky (2017), which makes our work applicable to a broad range of problems.

Second, The value of Eq.3 is bounded. The possible value of $\Phi(x_{i,\omega_i}, \hat{x}_{i,\omega_i}) = |I(\hat{x}_{i,\omega_i}; x_{i,\overline{\omega_i}}, y_i, f) - I(x_{i,\omega_i}; x_{i,\overline{\omega_i}}, y_i, f)|$ and $\Psi(x_{i,\omega_i}, x_{i,\overline{\omega_i}}) = |I(x_{i,\omega_i}; f) - I(x_{i,\overline{\omega_i}}, y_i; f)|$ follow the constraint that:

$$\begin{aligned}\Phi(x_{i,\omega_i}, \hat{x}_{i,\omega_i}) &\leq \log(q\epsilon) \\ \Psi(x_{i,\omega_i}, x_{i,\overline{\omega_i}}) &\leq I(z_i; f)\end{aligned} \tag{15}$$

where the maximum cardinality of any single feature in the feature subset $\omega_i$ is denoted as q. $\epsilon$ is the maximum number of features that the attacker may perturb, a.k.a the attack budget. the number of the features in $\omega_i$, noted as $|\omega_i|$ is no more than $\epsilon$. With this constraint, the value of Eq.3 is bounded from above as:

$$
\begin{aligned}
\mathcal{R}_f^{adv} - \hat{\mathcal{R}}_f^{adv} &\leq \frac{L\,\mathcal{D}_f}{\sqrt{2}n} \sqrt{\sum_{i=1}^{n} I(f; z_i) + 2\sum_{i=1}^{n} \Psi(x_{i,\omega_i}, x_{i,\overline{\omega_i}}) + \sum_{i=1}^{n} \Phi(x_{i,\omega_i}, \hat{x}_{i,\omega_i})} \\
&\leq \frac{L\,\mathcal{D}_f}{\sqrt{2}n} \sqrt{\sum_{i=1}^{n} 3I(f; z_i) + n\log(q\epsilon)}
\end{aligned}
\tag{16}
$$

In Eq.16, the first term under the squared root symbol is $\sum_{i=1}^{n} 3I(f; z_i)$. It measures the generalization error under the adversary-free setting according to Eq.13. The second term $\log(q\epsilon)$ measures the strength of the attack by considering the cardinality of the feature subset $\omega_i$. A higher cardinality $\log(q\epsilon)$ implies a larger combinatorial set of possible categorical feature values available to the attacker (more features that the attacker may perturb and/or more category values per feature that the attacker may choose to replace the original feature value). The attacker selects one set of categorical values in this combinatorial set to replace the original feature values within the feature subset $\omega_i$, in order to deliver the adversarial attack. Consequently, a higher cardinality indicates greater flexibility to organize feature manipulation over $\omega_i$, which signifies a stronger attack and thereby elevates the adversarial risk. Eq.16 gives a bounded but rough estimate of the adversarial risk, as not all of the features are useful for attack. Only the perturbation over influential features may cause effectively the rise of adversarial risk. In this sense, Eq.3 provides more accurate estimate to the actual adversarial risk than Eq.16.

## B    CONNECTION BETWEEN THE THEORETICAL ANALYSIS AND THE DESIGN OF IGSG

Our design of adversarially robust training is in accordance with two recommended factors to minimize the adversarial risk. However, deriving consistent and differentiable estimates of mutual information between high-dimensional variables, such as the parameters of deep neural networks and input categorical feature vectors, remains an open and challenging problem due to the curse of dimensionality Gao et al. (2018). This makes direct optimization of the mutual information-based bound impractical. To reach this goal, we propose the *IGSG*-based robust training paradigm. It jointly applies two smoothness-enhancing regularization techniques into the learning process of a classifier with categorical inputs, in order to mitigate the adversarial attack over categorical data. We discuss the design of IGSG in the followings. To further confirm the effectiveness of *IGSG* in minimizing the mutual-information-based adversarial risk bound, we provide approximated computation of the mutual-information based bound with the toy model in Appendix.I.1. We derive the estimated bound value derived with and without applying our proposed robust training mechanism. The empirical observations show that enforcing the two regularization terms indeed decreases the estimated value of the bound, which echoes the rise of adversarial accuracy.

**Minimizing $I(f; z_i)$ by smoothing the curvature of the classification boundary.**    In previous work, Fisher information $\rho(z_i)_f$ was utilized as a quantitative measure of the information that the hypothesis $f$ contains about the training sample $z_i$ (Hannun et al., 2021). As shown in Wei & Stocker (2016), $\rho(z_i)_f$ is closely related to the mutual information $I(f; z_i)$, higher/lower $\rho(z_i)_f$ indicates higher/lower $I(f; z_i)$. Our work aims to minimize $\rho(z_i)_f$ to effectively penalize excessively high mutual information $I(f; z_i)$. The computation of $\rho(z_i)_f$ is detailed in Eq.16 of (Hannun et al., 2021). In this context, suppressing $\rho(z_i)_f$ (approximately suppressing $I(f; z_i)$) is equivalent to penalizing the magnitude of the gradient of the loss function with respect to each $z_i$. This approach, supported by findings in (Smilkov et al., 2017), uses gradient regularization to smooth the classifier's decision boundary, thereby reducing the potential risk of overfitting and enhancing adversarial resilience

**Minimizing $\Psi(x_{i,\omega_i}, x_{i,\overline{\omega_i}})$ and $\Phi(x_{i,\omega_i}, \hat{x}_{i,\omega_i})$ via smoothing the distribution of feature-wise contribution to the classification output.** Minimizing these terms involves evaluating the mutual information between the feature subset $\omega_i$ and the combined set of remaining features and the trained

model $f$. Approximating this mutual information-based penalization with Fisher information is thus infeasible. The primary goal of regularizing these terms is to prevent the classifier from relying too heavily on a few influential features. To achieve this, we propose using Integrated Gradient (IG) (Sundararajan et al., 2017) to assess feature-wise contributions to the classification output. We apply Total-Variance (TV) regularization over the feature-wise Integrated Gradient to promote a smooth and balanced distribution of feature-wise attribution. In Appendix.I.1, we show empirically with toy models that performing the proposed TV regularization can reduce the estimated value of both mutual information-based terms.

## C   DIFFERENCE BETWEEN PAC-BAYES BOUNDS AND OUR STUDY

Following (Xu & Raginsky, 2017; Bu et al., 2019), we don't impose any prior distribution assumption over $P_{f|S^n}$. This characterizes the major difference between our study and PAC-Bayes generalization bounds (McAllester, 1999). Though PAC-Bayesian bounds also connect information-theoretic quantities to generalization and are similar to the mutual information approach, these bounds are usually output dependent–that is,they give a generalization bound for a particular output hypothesis or hypothesis distribution,rather than uniformly bounding the expected error of the algorithm as does in the mutual-information based bound in our study. We adopt the mutual-information based technique to exploit the fact that the generalization error depends strongly not only on the underlying true data-generating distribution, but also on the correlation between the collection of empirical risks of the available hypotheses and the final output of the learning algorithm.

## D   DISCUSSION ABOUT THE RANDOMIZED LEARNING MECHANISM

It is worth noting that our information-theoretic analysis roots in the research of mutual information based generalization error analysis in (Xu & Raginsky, 2017; Bu et al., 2019). This line of inquiry adopts an information-theoretic perspective to enhance the generalization capabilities of machine learning algorithms. Within this theoretical framework, a model training algorithm is conceptualized as a randomized mapping or an information-transmitting channel, employing the language of information theory. This mapping or channel takes a training dataset as input and yields a hypothesis as output. The randomness inherent in this mapping/channel manifests in two dimensions. First, the training dataset provided to the channel is a sample selected from all possible combinations of n training data points. Second, the resulting hypothesis from this channel is one sample chosen from the set of possible hypotheses within the hypothesis space. The mutual information-based bound in Eq.3 thus determine the expected adversarial risk over all possible hypothesis functions in the hypothesis space. In other words, we offer an averaged estimate of the potential adversarial risk, irrespective of the hypothesis chosen as the output by the learning algorithm. In this sense, for a classifier used in a concrete learning task, whether the parameters/decision outputs of this classifier are deterministic or randomized, our mutual-information-based bound is applicable.

## E   DETAILED DESIGN OF INTEGRATED GRADIENT AND SMOOTHED GRADIENT REGULARIZATION

**Definition 5.** *(**Total Variation of IG-based Regularization**). The objective function of the classifier $f$ with TV loss is defined as,*

$$\min_\theta \mathbb{E}_{(x_i,y_i)\sim\mu_z} \{\ell(x_i, y_i; \theta) + \alpha\ell_{TV} IG(x_i)\} \tag{17}$$

*where $\alpha$ is a hyper-parameter tuning the weight of the TV regularization term, and $\theta$ is the parameters of $f$. $\ell(x_i, y_i; \theta)$ is the learning loss of $f$, e.g. the cross entropy loss function. $\ell_{TV}(\cdot)$ denotes the TV loss of the IG scores of $x_i$. We follow the implementation of the TV loss over time series data, i.e. $\ell_{TV} IG(x_i) = \sum_{j=1}^{p-1} |IG(x_i)_j - IG(x_i)_{j+1}|$.*

In $\ell_{TV}(\cdot)$, we normalize the IG scores of each feature $x_{i,j}$ with softmax transformation. Therefore, the normalized IG score of each feature is valued within $[0, 1]$ and sums up to 1. By minimizing the regularization term based on TV loss, the distribution of the IG scores is driven to be as uniform as possible.

**Definition 6.** *(Smoothed Gradient Regularization).* *With $R$ randomly sampled data points $x_1, x_2, \ldots, x_R$ around the input instance $x_i$, the gradient smoothing-oriented regularization term defined on $x_i$ is given as follows:*

$$\min_{\theta} \mathbb{E}_{(x_i, y_i) \sim \mu_z} \ell(x_i, y_i; \theta) + \frac{\beta}{R} \sum_{r=1}^{R} ||G_r||_q \tag{18}$$

*where $\beta$ is a hyper-parameter, and $G_r \in \mathbb{R}^{p*m}$ is a gradient matrix with:*

$$G_{r,j,k} = g_{r,j,k} - g_{r,j,k^*} \tag{19}$$

*where $g_{r,j,k} = \frac{\partial}{\partial b(x_r)_{j,k}} \ell(x_r, y_i; \theta)$. We use $L_Q$ norm to calculate the norm of the gradient. Specifically, we choose q=2 for all the experiments following (Ross & Doshi-Velez, 2018b).*

We next elaborate on the details of the calculation in Eq.18 for categorical data, since it is different from the vanilla smoothed gradient computing with continuous input (Smilkov et al., 2017). **First**, we choose the categorical instances $\{x_r, r = 1...R\}$ by randomly changing a few features of $x_i$, such that $|diff(x_i, x_r)|$ equals to the attack budget $\epsilon$. By taking the gradients associated with $\{x_r, r = 1...R\}$ that are similar categorical vectors to $x_i$, we aim to obtain a more accurate measurement of the smoothness of the decision boundary around $x$. We average the magnitudes of the gradient vectors of $\{x_r, r = 1...R\}$ for each categorical instance $x_i$. Empirically, we choose $R = 5$, which brings consistently good results without very high time complexity. **Second**, instead of using the vanilla gradient, we inherit the idea of mean field smoothing as defined in Eq.10 and 13 of (Herault & Horaud, 1995) over the gradient values associated with each categorical feature of $x_r$. As shown in Eq.19, for each feature of $x_r$ (noted as $x_{r,j}$), we minimize the norm of the difference between $g_{r,j,k}$ and $g_{r,j,k^*}$, where $k^*$ denotes the category value carried by $x_{r,j}$. It is formulated as minimizing the $L_Q$ norm of the difference of gradients $G_r$ in Eq.18. By optimizing with the regularization term, our aims are two-fold: a) We suppress the magnitude of the gradient with respect to each categorical feature $x_{r,j}$ to reduce the adversarial risk. b) We smooth the distribution of the gradient values $g_{r,j,k}$ associated with the optional category values of each categorical feature $x_{r,i}$. Domain-agnostic discrete attacks, e.g., Orthogonal Matching Pursuit Greedy Search (OMPGS) (Wang et al., 2020b), rank the gradient values associated with the one-hot encoded vector $x_{r,i}$. The top-ranked category values other than $x_{r,i} = k^*$ are selected by OMPGS as the candidates of feasible adversarial perturbation to replace $x_{r,i} = k^*$. Minimizing the difference between $g_{r,j,k}$ and $g_{r,j,k^*}$ produces a set of uniformly distributed gradient values $g_{r,j,k}$. It prevents gradient-guided attack methods like OMPGS from identifying promising candidates for generating effective adversarial perturbation.

## F EMPIRICAL STUDY OF THE ROBUST OVERFITTING ISSUE

Let $P_{tr}$ and $P_{te}$ ($O_{tr}$ and $O_{te}$) denote the adversarial samples produced by the PGD-based attack $P$ (OMPGS-based attack $O$), which are used respectively for adversarial training ($tr$) and testing ($te$). The empirical evaluation of distribution gap is conducted by comparing the following 4 groups of Wassernstein distance scores.

**Wassernstein distance between in-distribution samples (WD$_{in}$):** We first measure the Wassernstein distance between samples within each of $P_{tr}$, $P_{te}$, $O_{tr}$ and $O_{te}$. For each set, we randomly shuffle twice the adversarial samples and select $90\%$ of the samples from the set as the probe and gallery set. We then compute the Wassernstein distance between the probe and gallery set. This process is repeated for 20 times. We record all the Wassernstein distance scores to measure the distribution gap between in-distribution adversarial samples within each set. WD$_{in}$ is considered as a baseline. We expect the Wassernstein distance scores between adversarial samples from different distributions (Out-Of-Distribution) to be significantly larger than the distance scores in WD$_{in}$.

**Wassernstein distance between the training and testing adversarial samples produced by the PGD-based method (WD$_{out}^{P}$):** For $P_{tr}$ and $P_{te}$, we randomly sample $90\%$ of the adversarial samples from each set and compute the Wassernstein distance between the selected subset from the training and testing set. We repeat this process for 20 times and obtain the Wassernstein distance scores to measure the distribution gap between the training and testing adversarial samples generated by the PGD-based method.

**Wassernstein distance between the training and testing adversarial samples produced by the OMPGS-based method ($WD_{out}^{O}$):** For $O_{tr}$ and $O_{te}$, we randomly sample 90% of the samples from each set and compute the Wassernstein distance between the two selected subsets. This process is repeated 20 times to obtain all the Wasserstein distance scores, assessing the distribution difference between training and testing adversarial samples generated by the OMPGS-based attack method.

Table 5: Average and standard (AVG) deviation (STD) of the Wassernstein distance scores

| Group of Wassernstein distance | AVG | STD |
|---|---|---|
| $WD_{in}$ | 0.06 | 0.003 |
| $WD_{out}^{P}$ | 0.05 | 0.001 |
| $WD_{out}^{O}$ | 0.12 | 0.002 |
| $WD_{out}^{PO}$ | 0.18 | 0.002 |

**Wassernstein distance between the training and testing adversarial samples produced by the PGD-based and OMPGS-based attack methods ($WD_{out}^{PO}$):** We conduct a cross-check in this part. We randomly sample 90% of the samples from $P_{tr}$ and $O_{te}$ respectively and compute the Wassernstein distance between the selected subset of adversarial samples from the two sets. The same distance computing operation is also conducted on the subsets from $O_{tr}$ and $P_{te}$. This process is repeated for 20 times and obtain the Wassernstein distance scores to assess the distribution difference between training and testing adversarial samples generated using different attack methods.

In Table.5, the averaged Wassernstein scores of $WD_{in}$ and $WD_{out}^{P}$ are the smallest among the four groups of distance values. Conversely, $WD_{out}^{PO}$ and $WD_{out}^{O}$ rank as the largest and second largest, respectively. Our findings can be summarized from two perspectives. First, we conduct a Mann-Whitney U test on the distance scores of $WD_{in}$ and $WD_{out}^{P}$. The test results indicate no significant difference between the distance scores in these two groups, yielding a p-value of 0.20. This suggests that the PGD-based method generates discrete adversarial samples with similar distributions for both training and testing. Consequently, the PGD-based adversarial training achieves high adversarial accuracy, as observed in Table.1. Second, we conduct

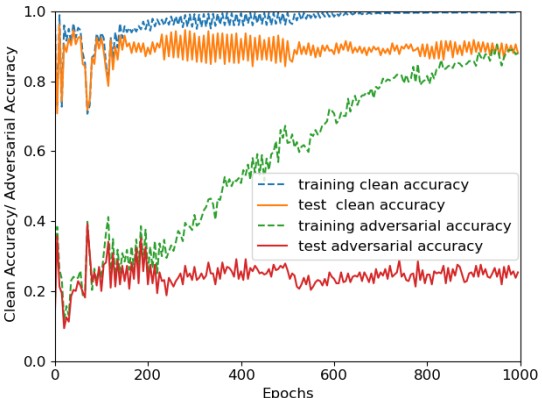

Figure 4: The "robust overfitting" of adversarially trained MLP on *Splice*.

Mann-Whitney U tests between $WD_{in}$ and $WD_{out}^{O}$, as well as between $WD_{in}$ and $WD_{out}^{PO}$. The hypothesis tests reveal that $WD_{out}^{O}$ and $WD_{out}^{PO}$ are significantly higher than $WD_{in}$, with p-values of 0.02 and 0.01, respectively. This indicates that 1) the training and testing adversarial samples generated by the OMPGS-based adversarial training method have different distributions and 2) the training adversarial samples generated by one method (either PGD-based or OMPGS-based) have a different distribution from the testing adversarial samples generated by the other method. These results align with the low adversarial accuracy of the PGD-based adversarial training method when facing the OMPGS-based attack, and vice versa. Additionally, the observations confirm the occurrence of robust overfitting in the OMPGS-based adversarial training method, as illustrated in Figure.4.

# G DISTINCTIVE FACTORS IN ROBUSTNESS WITH CATEGORICAL DATA

## G.1 DISTINCTIVE FACTORS IN ASSESSING ROBUSTNESS WITH CATEGORICAL DATA

We emphasize three critical distinctions in characterizing and evaluating the adversarial robustness of categorical data compared to continuous data. Firstly, categorical data exists in discrete space, where each feature represents a unique category. Adversarial manipulation of categorical features involves switching from one feasible category to another, rendering traditional $L_Q$ distance metrics inapplicable. Consequently, samples generated through PGD and FGSM attacks are considered infeasible to use over discrete data directly (Lei et al., 2019; Bao et al., 2021; Wang et al., 2020b). However, PGD adversarial training and TRADES are both applicable to relaxed categorical data.

Adversarial samples are generated by relaxing $b(x_i)$ into continuous data, yielding float categorical values in $\mathbb{R}^{p*m}$. While these samples are inappropriate for directly evaluating model robustness in the discrete domain, they are effective for adversarial training, fostering improved robustness, as discussed in the global response.

Secondly, attacking discrete data entails a complex NP-hard mixed-integer nonlinear programming challenge (Lee & Leyffer, 2011). Moreover, the volume of the adversarial space expands exponentially with the feature dimension. Although transitioning the discrete problem to the continuous domain yields approximate solutions, the intricate combinatorial nature impedes complete coverage of feasible discrete adversarial samples. Adversarial training relying on the relaxed solution to the discrete attack risks overfitting to these approximations. Our study confirms this limitation, where adversarial training struggles to significantly bolster the robustness of discrete data—especially in high-dimensional settings with substantial attack budgets.

Finally, it is essential to recognize that certifiable adversarial robustness and adversarial risk bounds established for the image domain do not hold for discrete data. These bounds are based on $L_Q$ distance ($q \geq 1$) and do not adequately explain the true factors influencing the adversarial risk of discrete data, as demonstrated in Theorem 1 of (Bao et al., 2021). Therefore, applying these bounds to discrete data would yield inaccurate and unreliable results.

## G.2 Distinctive Factors in $L_0$ robustness

Tsipras et al. (2018) demonstrated that a model relying on multiple weakly correlated features with the label can make high-confidence (low entropy) predictions, which appears to conflict with our proposed method for smoothing the impact of different features. However, Tsipras et al. (2018) primarily focused on the $L_Q$ attack scenario, where experiments involve $L_2$ and $L_\infty$ attacks. However, our focus is on enhancing the adversarial robustness of categorical data, When perturbing categorical features, the concept of "modification magnitude" loses relevance. Instead, each feature undergoes a transformation by switching between distinct category values (switching from its original category value to another one). In this context, evaluating robustness using $L_\infty$ attacks is infeasible, as mentioned in our earlier responses. Therefore, adversarial attacks on categorical data are framed within the $L_0$ attack framework, rather than the $L_\infty$ attack scenario. It's important to underline that distinct attack scenarios can yield varying conclusions regarding adversarial robustness. However, the fundamental concept driving adversarial robustness remains consistent for both $L_0$ and $L_Q$ attacks — mitigating overfitting on the training data is paramount.

For instance, in the context of $L_\infty$ attacks, overfitting often occurs with respect to the background. As every pixel can be perturbed to some extent, classifiers that overfit to background elements become susceptible to adversarial attacks. This concurs with the findings of (Tsipras et al., 2018). Standard models that utilize all features tend to be vulnerable, while adversarially trained models tend to focus on influential features. This vulnerability arises from the classifier's overfitting to background features. This leads us to the insight that due to the permissible perturbation of any feature within certain bounds, changing influential features to alternative patterns is notably more challenging than altering background features, thus rendering background overfitting a significant adversarial vulnerability .

Nonetheless, in the context of an $L_0$ norm bounded attack, the scenario differs. When weakly correlated features are perturbed, highly influential features still remain untouched within the confines of the $L_0$ norm constraint. Consequently, targeting the most influential features becomes a pathway to a successful attack, which is a contrast to the $L_\infty$ attack situation. As an echo, our defense thus aims to smooth the feature-wise contribution to the classifier, making the adversary difficult to identify influential features. This fundamental discrepancy is at the root of the disparities between our findings and those presented in (Tsipras et al., 2018).

## H    Detailed Experimental Settings

### H.1    The settings of FSGS and OMPGS

To evaluate adversarial robustness, we employ the FSGS attack and OMPGS attack, shown in Algorithm.1 and 2. The definition of the notations can be found in Appendix.H.4. It's also worth noting

---

**Algorithm 1** FSGS for general categorical data

---

**Input:** The candidate set $H = \{1, 2, ...p\}$ of all categorical features, categorical attack budget $\epsilon$

1:   $S \leftarrow \emptyset$
2:   **for** $iter = 0, 1, 2, \ldots$ **do**
3:     **for** each $j \in H/S$ **do**
4:       **for** each $s \subset S$, if $|s| < \epsilon$ **do**
5:         $\hat{x}(j, s) = B(x, \{j\} \cup s)$
6:       **end for**
7:       $m_f(x(j)) = \max\limits_{s \subset S, |s| < \epsilon} m_f(\hat{x}(j, s))$
8:     **end for**
9:     $m_f(x, S) = \max\limits_{j \in H/S} m_f(x(j))$
10:    $j^* = \arg\max\limits_{j \in H/S} m_f(x(j))$
11:    $S \leftarrow S \cup \{j^*\}$
12:    **if** $m_f(x, S) \geq 0$ **then attack successfully**
13:    **if** $Time \geq \Gamma$ **then timeout**
14: **end for**

---

**Algorithm 2** OMPGS for general categorical data

---

**Input:** The candidate set $H = \{1, 2, ...p\}$ of all categorical features, categorical attack budget $\epsilon$

1:   $S \leftarrow \emptyset$
2:   **for** $iter = 0, 1, 2, \ldots$ **do**
3:     **for** $s \subset S$, if $|s| \leq \epsilon$ **do**
4:       $r_s \leftarrow \nabla f_y(B(x, s))$
5:       **if** $m_f(B(x, s)) \geq 0$
        **then attack successfully**
6:     **end for**
7:     **for** $j \in H/S$ **do**
8:       $s_j = \arg\max\limits_{s_j \subset S, |s_j| < \epsilon} |r_s[j]|,\ \hat{x}_j = B(x, \{j\} \cup s_j)$
9:     **end for**
10:    $j^* \leftarrow \arg\max\limits_{j \in H/S} m_f(\hat{x}(j))$
11:    $S \leftarrow S \cup \{j^*\}$
12:    **if** $Time \geq \Gamma$ **then timeout**
13: **end for**

---

that, in terms of attack methods for discrete data, while FSGS is a black-box attack and OMPGS is white-box, FSGS, with an extensive search, often encompasses the search space of OMPGS under the same attack budget, yielding higher success rates, as demonstrated in (Bao et al., 2021). For both methods, we impose a time constraint on each dataset. Specifically, we allocate 1s, 150s, and 2s for FSGS, and 1s, 5s, and 1.2s for OMPGS, corresponding to *Splice*, *PEDec*, and *Census* datasets, respectively. Adversarial accuracy, which measures the prediction accuracy on adversarial samples generated by FSGS or OMPGS, is used as the metric for assessing robustness. These settings are consistently applied to all methods, including *IGSG*, the baseline methods, and the ablation methods. In the case of mixed-type datasets like *Census*, we devise variations of FSGS and OMPGS to enhance the effectiveness of the attack. Further details can be found in Appendix.H.4.

### H.2 DETAILS OF THE BASELINE METHODS

1. Standard Training (*Std Train*) is the model trained with adversary-free data by cross-entropy.
2. PGD Adversarial Training (*Adv Train*) is the vanilla adversarial training (Madry et al., 2017).
3. *Fast-BAT* (Zhang et al., 2022) advances vanilla adversarial training from the perspective of bi-level optimization. It achieves a better accuracy-robustness balance than *Adv Train*.
4. *TRADES* (Zhang et al., 2019) optimizes a regularized surrogate loss composed of empirical risk minimization and a robustness regularization term.

5. Adversarial Feature Desensitization (*AFD*) (Bashivan et al., 2021) improves robustness by learning a feature space where the adversary-free and adversarial instances share the same distribution.

6. Probabilistic Adversarial Training (PAdvT) (Xu et al., 2023) first use Probabilistic Categorical Adversarial Attack (PCAA) proposed in the same paper to generate adversarial samples in discrete space and then uses these adversarial samples for adversarial training.

7. Input Gradient Regularization (*IGR*) (Ross & Doshi-Velez, 2018b) penalizes the magnitude of the vanilla gradient of the classification loss with respect to the training data.

8. Jacobian Regularization (*JR*) (Hoffman et al., 2019) proposes to penalize the approximation of the Frobenius norm of the Jacobian matrix.

The last seven baselines except the sixth baseline are all originally designed for continuous input. We relax the one-hot encoded representation of categorical training data when adapting these baselines to our test. For four adversarial training baselines (*Adv Train*, *Fast-BAT*, *TRADES* and *AFD*), we adopt $L_1$-norm bounded adversary in the inner maximization of the adversarial training process. When a mixture of categorical and numerical features presents (e.g., in *Census* dataset), the PGD-1 attack is applied for the categorical features and the PGD-$\infty$ attack is used for numerical features. For two regularization-based baselines (*IGR* and *JR*), we compute the gradient of the classifier's output (*JR*) / the classification loss (*IGR*) with respect to the continuous relaxation of the categorical data. The details about the hyper-parameters during training can be found in Appendix.H.3.

### H.3 THE SETTINGS OF THE HYPER-PARAMETERS IN THE TRAINING PHASE

First, we talk about the learning rate. We experiment with different learning rates for the MLP model. Specifically, we set the learning rates to 0.07, 0.2, and 0.008 for *Splice*, *PEDec*, and *Census* datasets, respectively. All methods utilizing IG regularization achieve the best performance using the same learning rate. For other methods, unless otherwise specified, we use learning rates of 0.07, 0.00001, and 0.008 to achieve optimal performance for the MLP model. In the case of *PEDec* using the IG-based training paradigm, we use a larger learning rate to achieve optimal solutions of the smoothness of IG scores for each feature. It is important to note that large learning rates would decrease both robustness and accuracy in other situations. For the Transformer model, we adopt learning rates of 0.003, 0.002, and 0.02 for *Splice*, *PEDec*, and *Census*, respectively."

When tuning the hyper-parameters $\alpha$ and $\beta$ of the proposed *IGSG* method in Eq.5, we analyze their sensitivity by testing different parameter values ranging from 0.01 to 100. We employ 10-fold cross-validation and evaluate the robustness using the OMPGS attack. In detail, we randomly and evenly divide the training set into 10 parts. Each time, we use one part as test set and others as training set. We train an MLP classifier with varied $\alpha$ and $\beta$. We do the whole process for 10 times and each part is regarded as the test set for once. After that, we calculate the average of the adversarial accuracy under OMPGS attack for each setting of the hyper parameters. Figure 5 illustrates the adversarial robustness of the MLP model for *Splice* and *PEDec* datasets. For *Splice*, we consistently obtain excellent results, as different combinations of $\alpha$ and $\beta$ have small impact on the adversarial accuracy. However, for *PEDec*, we observe that the left side of the box consistently performed well. When using a small $\alpha$ value, good results are achieved regardless of the choice of $\beta$. Hence, when applying the *IGSG* method, it is unnecessary to exhaustively explore all combinations of $\alpha$ and $\beta$. Balancing the three parts of the loss function typically leads to satisfactory performance.

Confidence intervals are calculated to gauge the reliability of the adversarial accuracy obtained through cross-validation. For *PEDec*, the length of the confidence interval ranges from 0.1 to 0.15 at a 95% confidence level. Conversely, for *Splice*, the interval is approximately 0.1 at a 95% confidence level. In the case of the MLP model, we choose $\alpha$ values of 10, 0.01, and 1 for the three datasets, respectively. As for $\beta$, we use values of 100, 0.1, and 3 for the respective datasets. For the Transformer model, $\alpha$ is set to 100 for all three datasets, while $\beta$ takes values of 1, 0.1, and 1 for the respective datasets.

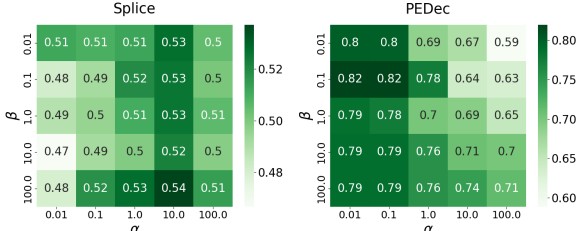

Figure 5: Adversarial Accuracy of *IGSG* under different $\alpha$ and $\beta$ of the MLP model

In the case of the PGD-1 attack in the *Adv Train*, *AFD*, and *TRADES* methods, we set $\epsilon$ to be 5 for the three datasets. The attack consists of 20 iterations, with the attack step size set to $\epsilon/10$. Regarding Fast-BAT (Zhang et al., 2022), we also set $\epsilon$ to be 5 for the three datasets. The attack step size is determined as $\epsilon/4$.

In the *IGR* method, the parameter that weighs the importance of the norm of the input gradient is set to the same value as $\beta$ in Eq.5. For the MLP model, we use the values of 100, 0.1, and 3 for the three datasets, respectively. As for the Transformer model, the values are set as 1, 0.1, and 1 for the respective datasets.

In the *JR* method, the hyper-parameter that weighs the importance of the Frobenius norm of the Jacobian matrix is tuned to achieve optimal robustness. For the MLP model, we set the values of 0.5, 1, and 0.02 for the three datasets, respectively. As for the Transformer model, the values are set as 1, 0.05, and 0.1 for the respective datasets.

In the *AFD* method, Algorithm 1 in (Bashivan et al., 2021) includes three learning rates. For the MLP model, we set the values of $\alpha$ to be 0.01, 0.00001, and 0.008, $\beta$ to be 0.001, 0.0005, and 0.0001, and $\gamma$ to be 0.001, 0.00005, and 0.0001 for the three datasets, respectively. As for the Transformer model, we set $\alpha$ to be 0.001, 0.002, and 0.0001, $\beta$ to be 0.001, 0.0001, and 0.001, and $\gamma$ to be 0.001, 0.0001, and 0.0001 for the three datasets, respectively.

In the *TRADES* method described in (Zhang et al., 2019), we set the parameter $\lambda$ to balance accuracy and robustness. Specifically, for the MLP model, we set $\lambda = 1$ for the *Splice* and *Census* datasets, and $\lambda = 0.2$ for the *PEDec* dataset. As for the Transformer model, we set $\lambda = 1$ for the all the three datasets.

In Eq.13 of the *Fast-BAT* method (Zhang et al., 2022), we set the values of the parameters as follows: $\alpha_1 = \epsilon/4$, $\lambda = 1/\alpha_1$, $\alpha_2 = 1$ for the *Splice* dataset, and $\alpha_2 = 0.1$ for the *PEDec* and *Census* datasets.

For the training epochs, we execute 3000, 180, and 100 epochs on *Splice*, *PEDec*, and *Census* respectively. We perform 5 runs of all the methods and computed the average score and standard deviation. When evaluating the adversarial accuracy under OMPGS attack of different methods on different attack budgets, we pick the best one among the 5 runs for each method to draw Figure.2 and Figure.8.

### H.4 SPECIAL SETTINGS FOR MIXED-TYPE DATASET

For mixed-type datasets that contain both categorical and numerical features, direct application of FSGS, OMPGS, or PGD attacks is not suitable for evaluating the robustness of the classifier. This is because categorical data requires an $L_0$ attack, while numerical data typically necessitates an $L_2$ or $L_\infty$ attack.

To address this challenge and evaluate the adversarial robustness of a mixed-type classifier, an iterative approach is employed. This approach involves running FSGS or OMPGS along with PGD attacks iteratively to obtain a more effective adversary. This combination allows for a comprehensive evaluation of the robustness of the mixed-type classifier.

Before talking about the details, we note that there are $p_{cat}$ categorical features and $p_{num}$ numerical features. Each categorical feature has $m$ candidate values. For a sample $x$, the value of feature $j$ is $k^*$. After perturbation, the value is $\hat{k}$. The ground truth label of $x$ is $y^*$. During the attack, we maintain a greedy set $S$, showing the alterable features. Each feature not in $S$ cannot be changed, i.e. for $j \notin S$, $\hat{k} = k^*$. For the features in $S$, it is possible to choose any of the $m$ candidate values, and it is also acceptable to remain unchanged. Here we introduce the notation in (Bao et al., 2021). Given a greedy set $S$,

$$m_f(x) = \max_{y \neq y^*}\{f_y(x)\} - f_{y^*}(x)$$

$$m_f(x, S) = \max_{diff(x,\hat{x}) \subset S} m_f(\hat{x})$$

where we denote $diff(x, \hat{x})$ as the set of feature indices where $\hat{k} \neq k^*$. The function $m_f(x)$ indicates whether the sample $x$ is misclassified. If $m_f(x) < 0$, it means that $x$ is classified correctly, while $m_f(x) \geq 0$ indicates misclassification. The function $m_f(x, S)$ checks whether the attack is

---

**Algorithm 3** FSGS + PGD for mixed-type data

---

**Input:** The candidate set $H = \{1, 2, ...p_{cat}\}$ of all categorical features, PGD attack budget $\epsilon_n$ for numerical data, categorical attack budget $\epsilon_c$

1: $S \leftarrow \emptyset$
2: **for** $iter = 0, 1, 2, \ldots$ **do**
3:     **for** each $j \in H/S$ **do**
4:         **for** each $s \subset S$, if $|s| < \epsilon_c$ **do**
5:             $\hat{x}(j, s) = B(x, \{j\} \cup s)$
6:             $\delta(j, s) = \text{PGD}_\infty(\hat{x}(j, s), \epsilon_n)$
7:             $\hat{x}(j, s) = \hat{x}(j, s) + \delta(j, s)$
8:         **end for**
9:         $m_f(x(j) + \delta(j, S)) = \max\limits_{s \subset S, |s| < \epsilon_c} m_f(\hat{x}(j, s))$
10:     **end for**
11:     $m_f(x + \delta, S) = \max\limits_{j \in H/S} m_f(x(j) + \delta(j, S))$
12:     $j^* = \arg\max\limits_{j \in H/S} m_f(x(j) + \delta(j, S))$
13:     $S \leftarrow S \cup \{j^*\}$
14:     **if** $m_f(x + \delta, S) \geq 0$ **then attack successfully**
15:     **if** $Time \geq \Gamma$ **then timeout**
16: **end for**

---

**Algorithm 4** OMPGS + PGD for mixed-type data

---

**Input:** The candidate set $H = \{1, 2, ...p_{cat}\}$ of all categorical features, PGD attack budget $\epsilon_n$ for numerical data, categorical attack budget $\epsilon_c$

1: $S \leftarrow \emptyset$
2: **for** $iter = 0, 1, 2, \ldots$ **do**
3:     **for** $s \subset S$, if $|s| \leq \epsilon_c$ **do**
4:         $r_s \leftarrow \nabla f_y(B(x, s))$
5:         **if** $m_f(B(x, s) + \text{PGD}_\infty(B(x, s), \epsilon_n)) \geq 0$
        **then attack successfully**
6:     **end for**
7:     **for** $j \in H/S$ **do**
8:         $s_j = \arg\max\limits_{s_j \subset S, |s_j| < \epsilon_c} |r_s[j]|, \; \hat{x}_j = B(x, \{j\} \cup s_j)$
9:     **end for**
10:     $j^* \leftarrow \arg\max\limits_{j \in H/S} m_f(\hat{x}(j) + \text{PGD}_\infty(\hat{x}(j), \epsilon_n))$
11:     $S \leftarrow S \cup \{j^*\}$
12:     **if** $Time \geq \Gamma$ **then timeout**
13: **end for**

---

successful under the constraints of the feature set $S$. The notation $B(x, s)$ represents the adversarial sample $\hat{x}$ obtained by modifying the features of $x$ as indicated by the binary vector $s$. Algorithm.3 outlines the attack process using FSGS+PGD for mixed-type data, while Algorithm.4 describes the attack process using OMPGS+PGD for mixed-type data. For general categorical data where there are no numerical features, the "PGD" step in the algorithms can be ignored or $\epsilon_n$ can be set to 0.

During the experiment, each feature is normalized before applying the PGD attack. For PGD-$\infty$ attack, we set $\epsilon_n = 0.2$ for the *Census* dataset, with a total of 20 attack steps. The attack step size is set to 0.02. During the training process of *Adv Train*, *AFD*, *TRADES*, and *Fast-BAT*, we use a combination of PGD-1 attack for categorical features and PGD-$\infty$ attack for numerical features to generate adversarial samples for mixed-type data. The same attack settings are applied during the training of *Adv Train*, *AFD*, and *TRADES*. For *Fast-BAT*, we also set $\epsilon_n = 0.2$, but the attack step size is adjusted to 0.05.

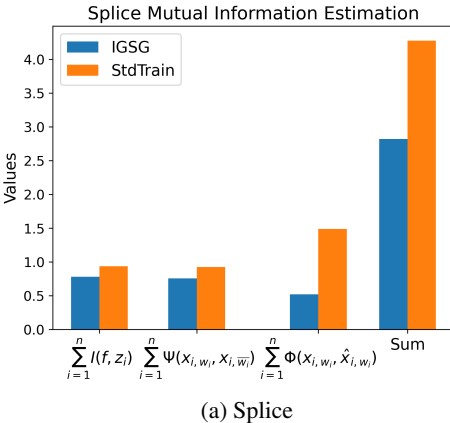
(a) Splice

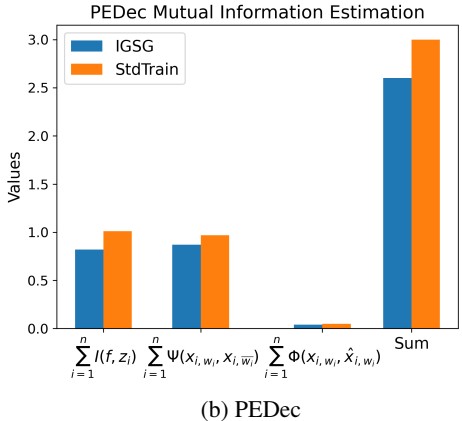
(b) PEDec

Figure 6: Mutual Information Estimation for terms in Eq.3 for *Splice* and *PEDec* Datasets

# I  ADDITIONAL EXPERIMENTAL RESULTS

## I.1  APPROXIMATION TO THE MUTUAL INFORMATION-BASED ADVERSARIAL RISK BOUND

In this section, we evaluate the mutual information as delineated in the adversarial risk bound (Eq.3), comparing models trained via Std Train and IGSG methods. Given the intricacies and potential inaccuracies in assessing an entire neural network, we focus on a simplified model comprising a single fully connected layer, with softmax activation for multi-class classification and sigmoid activation for binary classification. We utilize the Mutual Information Neural Estimation (MINE) technique (Belghazi et al., 2018) to assess the terms and their weighted sum in Eq.3 .

For training, we randomly selected 200 and 500 samples, 20 times each, from the training sets of *Splice* and *PEDec* datasets, respectively. These samples undergo training using Std Train and IGSG approaches, with a learning rate of 0.001, over 200 and 1000 epochs, respectively. This process yields an approximate accuracy of 0.9 for both datasets. Subsequently, we evaluate the adversarial robustness of 20 models each from Std Train and IGSG, employing FSGS and OMPGS attacks. Regarding the most sensitive features $\omega_i$ in Eq.3, we predetermine them based on the top 5 features exhibiting the highest attack frequency in Std Train models on MLP under OMPGS attacks. These features were fixed across all samples. For *Splice*, $\omega_i$ are [28, 29, 30, 31, 32], and for *PEDec*, [3592, 3755, 3808, 4390, 4918]. Using these predetermined $\omega_i$, we calculate the four mutual information terms, as illustrated in Figure 6, based on the 20 sampled datasets and corresponding model parameters, utilizing the MINE methodology. We also calculate the average adversarial accuracy on FSGS and OMPGS, the reuslt is shown in Table 6.

This experiment aims to demonstrate two key aspects. Firstly, IGSG-trained networks exhibit a reduction in the mutual information terms in Eq.3, suggesting a lower adversarial risk bound. Secondly, beyond just a lower adversarial risk bound, IGSG-trained networks also empirically manifest enhanced adversarial accuracy.

The results displayed in Figure 6 encompass four mutual information terms related to the adversarial risk bound. We first examine "Sum". "Sum" is defined as $\sum_{i=1}^{n} I(f; z_i) + 2\sum_{i=1}^{n} \Psi(x_{i,\omega_i}, x_{i,\overline{\omega_i}}) + \sum_{i=1}^{n} \Phi(x_{i,\omega_i}, \hat{x}_{i,\omega_i})$, representing the adversarial risk bound in Eq.3. We can refer to Table 6 for the average adversarial accuracy across 20 models trained on randomly sampled data under FSGS and OMPGS attacks. For both

Table 6: Average Adversarial Accuracy on 20 logistic regression models for *PEDec* and *Splice* datasets.

| Dataset | Attack | IGSG | Std Train |
|---------|--------|-------|-----------|
| Splice | FSGS | 0.019 | 0.010 |
| | OMPGS | 0.139 | 0.122 |
| PEDec | FSGS | 0.709 | 0.648 |
| | OMPGS | 0.748 | 0.668 |

*Splice* and *PEDec* datasets, the IGSG method typically yields lower "Sum" values and higher adversarial accuracy, corroborating that IGSG effectively reduces the adversarial risk bound in Eq.3 and that this reduction positively correlates with improved adversarial accuracy.

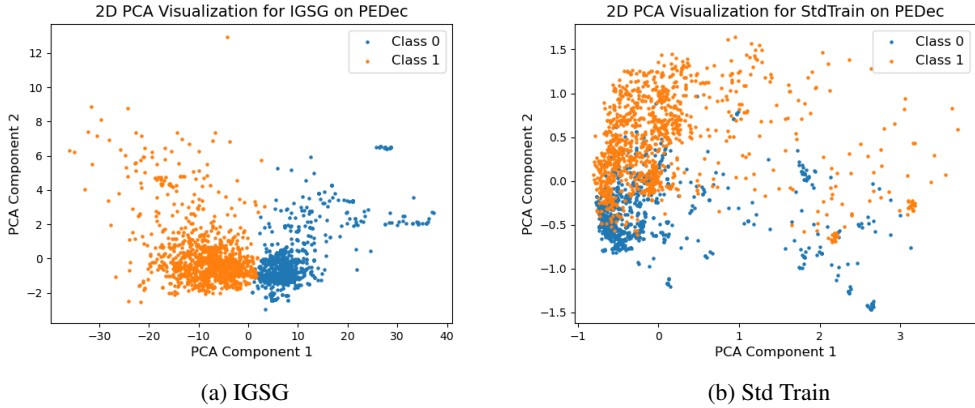

(a) IGSG  (b) Std Train

Figure 7: 2D PCA Boundary Visualization on *PEDec* Dataset

Focusing on the first three terms in Figure 6, we observe that $\sum_{i=1}^{n} I(f, z_i)$, indicative of adversary-free generalization error, is lower after using SG regularization compared to Std Train, signifying a more generalized classifier. The term $\sum_{i=1}^{n} \Psi(x_{i,\omega_i}, x_{i,\overline{\omega_i}})$ quantifies the differential contribution of highly vulnerable features $\omega_i$ and other features $\overline{\omega_i}$. Here, classifiers trained with IGSG typically exhibit lower values, suggesting a more balanced reliance on diverse features. For $\sum_{i=1}^{n} \Phi(x_{i,\omega_i}, \hat{x}_{i,\omega_i})$, which measures the sensitivity of the most vulnerable features $\omega_i$ to adversarial perturbations, IGSG-trained classifiers generally show lower values, particularly in the *Splice* dataset. This trend is attributed to the high vulnerability of certain features in $\omega_i$ for *Splice*, as evident in Figure 3. Perturbations in a single feature often lead to significant drops in prediction scores, resulting in larger values for Std Train, while IGSG effectively reduces this effect. For *PEDec*, successful attacks are usually driven by a combinatorial search. The combination of features with high attack frequency does not necessarily lead to successful attacks, hence the lower values for both Std Train and IGSG in this term. In summary, we observe that the classifier trained with IGSG exhibits lower values for all the four mutual information terms in the proposed upper bound in Eq.3 (thus a globally lower bound value) and higher adversarial accuracy across the two datasets. This finding firstly indicates that enforcing IGSG regularization can reduce the mutual information based upper bound of the adversarial risk proposed in Eq.3. Furthermore, we consider adversarial accuracy as a measure of actual adversarial risk. Higher adversarial accuracy indicates lower adversarial risk and vice versa. This quantitative evaluation demonstrates the correlation between the upper bound and actual adversarial risk. Lower values of the mutual information bound signify higher adversarial accuracy, thus indicating a reduced level of adversarial risk.

### I.2 VISUALIZATION OF THE CLASSIFICATION BOUNDARIES

In this section, we present a visualization of classification boundaries for classifiers trained using IGSG and Std Train methods, specifically for the *PEDec* dataset. We employ Multi-Layer Perceptron (MLP) classifiers trained via both IGSG and Std Train approaches. The visualization focuses on the features preceding the final fully connected layer within the test set. These features are compressed into a 2-dimensional space using Principal Component Analysis (PCA) for clearer representation.

Each sample in this visualization is labeled according to its predicted class by each respective classifier, offering an intuitive depiction of the classification boundaries. The results, as illustrated in Figure.7, reveal distinct differences between the two training methodologies. The IGSG-trained classifier exhibits an almost linear and distinct boundary between the two classes in the PCA visualization. In contrast, the Std Train-trained classifier's visualization does not present a clear demarcation. There is considerable overlap between the two classes in the PCA visualization of features from the last layer, indicating a twisted classification boundary.

Table 7: Adversarial Accuracy under FSGS attack and Accuracy (%) for *IGSG* and baseline models for the Transformer model. Adv Train (Madry et al., 2017), Fast-BAT (Zhang et al., 2022), TRADES (Zhang et al., 2019), AFD (Bashivan et al., 2021), PAdvT (Xu et al., 2023), IGR (Ross & Doshi-Velez, 2018b), JR (Hoffman et al., 2019)

| Dataset | Attack | Undefended Std Train | Adversarial Training baselines | | | | | Regularization baselines | | Ours |
|---------|--------|------|----------|----------|--------|--------|--------|--------|--------|--------|
| | | | Adv Train | Fast-BAT | TRADES | AFD | PAdvT | IGR | JR | **IGSG** |
| Splice | budget=5 | 0.9±0.9 | 0.4±0.5 | 1.0±1.1 | 0.0±0.0 | 0.2±0.4 | 0.2±0.4 | 0.4±0.3 | 0.1±0.1 | **2.3±1.4** |
| | Clean | 96.9±0.4 | 96.7±0.8 | 96.4±0.5 | 96.2±0.6 | 93.7±1.5 | 95.6±0.6 | 96.4±0.2 | 92.9±1.7 | 96.7±0.7 |
| PEDec | budget=5 | 41.1±4.1 | 60.6±0.7 | 49.9±3.8 | 59.0±3.9 | 48.1±9.8 | 22.6±1.3 | 59.5±5.0 | 62.2±1.9 | **63.5±3.7** |
| | Clean | 96.2±0.5 | 96.0±0.2 | 96.1±0.1 | 96.7±0.1 | 96.1±0.4 | 96.2±0.1 | 95.5±0.1 | 93.1±1.8 | 95.7±0.3 |
| Census | budget=5 | 27.6±4.3 | 34.1±2.7 | 33.1±6.1 | 32.2±8.0 | 32.2±1.0 | 30.4±3.4 | 25.1±5.3 | 32.7±0.4 | **37.8±4.3** |
| | Clean | 95.2±0.1 | 95.2±0.1 | 93.4±1.1 | 94.4±0.1 | 95.1±0.0 | 95.1±0.0 | 95.1±0.1 | 94.9±0.2 | 94.8±0.1 |

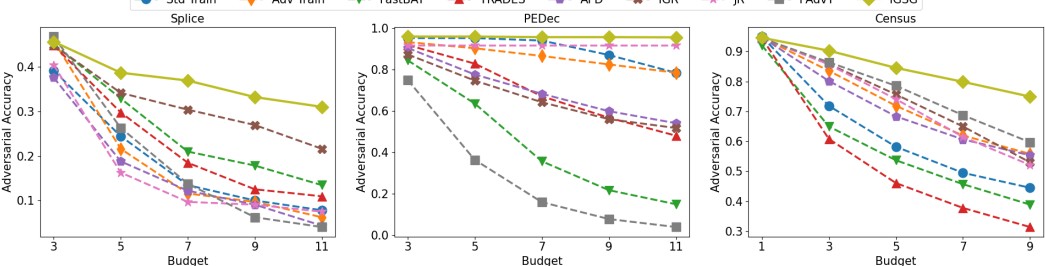

Figure 8: Adversarial accuracy for *IGSG* and baselines under OMPGS attack with varied attack budgets for the Transformer model.

This observation underscores that, compared to Std Train, IGSG facilitates a smoother and more discernible classification boundary. Such a visualization not only highlights the distinctiveness of the IGSG method but also demonstrates its efficacy in achieving clearer class separations.

## I.3 EXPERIMENTAL RESULTS ON TRANSFORMER MODELS

In addition to implementing *IGSG* on the MLP model to demonstrate its effectiveness, we also conducted experiments on a Transformer model. Table 7 presents the accuracy and adversarial accuracy against FSGS attack for each robust training method used with the Transformer model. For the *Splice* dataset, we observed that none of the methods provided effective defense for the Transformer model. This could be attributed to the presence of particularly sensitive features in the *Splice* dataset, as mentioned in Section.5.2. The Transformer model amplifies this effect by focusing more attention on these features, resulting in lower adversarial accuracy. However, *IGSG* achieved comparatively higher adversarial accuracy. Regarding the *PEDec* dataset, *IGSG* demonstrated slight improvement compared to other methods, and the differences in adversarial robustness among the different robust training methods were not significant. This may be due to the self-attention layer in the Transformer model, which makes the relationships between different features less flexible compared to the MLP model. For the *Census* dataset, most of the baseline methods did not exhibit substantial improvement over the baseline model. However, *IGSG* showed a significant improvement of 10.2% compared to the undefended model.

In Figure 8, we present the adversarial accuracy of all the methods when subjected to OMPGS attacks with varying budgets for the Transformer model. As discussed in Section.5.2, higher adversarial accuracy and a lower decrease rate of adversarial accuracy with increasing attack budgets indicate better model robustness. Similar to the results obtained with the MLP model, we observe that *IGSG* outperforms the baseline models in terms of adversarial accuracy under OMPGS attacks. Specifically, for the *Splice* dataset, *IGSG* exhibits a noticeably lower decrease rate of adversarial accuracy, although its adversarial accuracy is similar to some baseline methods when the attack budget is small. For *PEDEC*, most methods demonstrate very high adversarial accuracy compared to the MLP model. This may be because the multi-head paradigm in the self-attention layer makes the gradient less informative compared to the MLP model. In this scenario, *IGSG* achieves the highest adversarial accuracy, with almost no samples successfully attacked as the attack budget increases. The *JR* method also maintains a constant adversarial accuracy as the attack budgets increase, but it is susceptible to attacks on a few samples when the budget is small and its accuracy is inferior

Table 8: Adversarial Accuracy and Accuracy over clean samples (%) for *IGSG* and other methods alleviating robust overfitting.

| Dataset | Adversary | Adv Train | KD+SWA | MLCAT$_{LS}$ | MLCAT$_{WP}$ | IGSG |
|---------|-----------|-----------|--------|--------------|--------------|------|
| Splice | FSGS | 43.6±0.7 | 36.8±1.9 | 25.4±2.8 | 24.6±1.7 | **44.0±2.6** |
| | OMPGS | 51.7±1.4 | 41.2±2.3 | 30.3±2.7 | 29.9±2.0 | **63.8±4.2** |
| | Clean | 96.2±0.4 | 94.0±1.5 | 94.4±0.8 | 94.6±1.2 | 95.9±0.7 |
| PEDec | FSGS | 53.1±1.7 | 62.5±3.5 | 45.8±3.2 | 52.8±4.5 | **86.5±3.8** |
| | OMPGS | 74.1±2.1 | 80.2±2.0 | 67.9±2.4 | 68.8±4.7 | **88.0±4.0** |
| | Clean | 96.2±0.0 | 96.6±0.1 | 96.8±0.1 | 95.3±0.2 | 95.5±0.2 |
| Census | FSGS | 54.1±2.3 | 65.4±4.4 | 53.2±3.7 | 52.6±2.9 | **67.2±3.5** |
| | OMPGS | 62.7±3.3 | 66.5±5.6 | 67.5±1.9 | 66.5±3.5 | **71.3±9.0** |
| | Clean | 94.5±0.3 | 95.3±0.1 | 94.6±0.0 | 94.8±0.2 | 95.5±0.2 |

to *IGSG*. Regarding the *Census* dataset, we observe that nearly all methods achieve an adversarial accuracy above 0.9 when modifying a single feature. As the attack budget increases, *IGSG* exhibits a significantly lower decrease rate compared to other methods.

### I.4 COMPARISON TO METHODS TARGETING AT ROBUST OVERFITTING

In this section, we give a comparison of the adversarial robustness between *IGSG* and proposed methods aiming to address robust overfitting. We consider two works in this comparison. (Yu et al., 2022) found that small-loss adversarial samples are the cause of robust overfitting. MLCAT was proposed to constrain the minimum loss. Loss scaling and weight perturbation are used for two implementation, denoted as MLCAT$_{LS}$ and MLCAT$_{WP}$ respectively. (Chen et al., 2020) used learned smoothing to mitigate robust overfitting. It introduced knowledge distillation to smooth the logits, and performed stochastic weight averaging to smooth the weights (denoted as KD+SWA). We implement these two works on the original PGD adversarial training (Adv Train (Madry et al., 2017)). The results are shown in Table.8. We can observe that IGSG consistently outperforms both of the two methods when alleviating the robust overfitting issue on categorical data. Also, KD+SWA has better performance than Adv Train on *PEDec* and *Census* datasets, but is inferior on *Splice* dataset. However, MLCAT is inferior to Adv Train under both LS and WP implementations. This may demonstrate that the statement that small-loss data cause robust overfitting may not be correct in categorical domain.

### I.5 TIME COMPLEXITY ANALYSIS

In this section, we give the time complexity of *IGSG* and compare the training time of *IGSG* with other baseline methods. Suppose $T$ in Eq.4 is the number of steps in the Riemman approximation of the integral in Integrated Gradient, $R$ in Eq.18 is the number of randomly sampled neighbors for each data point and $N$ is the number of samples in the training set. The time complexity for each iteration is thus $O(N * (T + R + 1))$. In comparison, OMPGS-based adversarial training has a complexity of $O(N * (2^\kappa + p * \kappa))$ for each iteration, where $\kappa$ represents the number of iterations within each attack and $p$ is the number of features.

We also measure the runtime cost of *IGSG* with the other baselines in Table.9, based on our implementation using the Python library PyTorch and conducting all the experiments on Linux server with a single GPU (NVIDIA V100). On *Splice*, *IGSG* requires significantly less training time compared to some adversarial training methods like *Adv Train*, *AFD* and *TRADES*. On *PEDec*, *IGSG* requires similar run-time, compared

Table 9: Time cost (min) for the training process for *IGSG* and baseline methods.

| Model | MLP | | | Transformer | | |
|-------|-----|--------|--------|-------------|--------|--------|
| Dataset | Splice | PEDec | Census | Splice | PEDec | Census |
| Std Train | 6 | 8 | 12 | 17 | 9 | 7 |
| Adv Train | 78 | 112 | 84 | 223 | 74 | 130 |
| Fast-BAT | 27 | 40 | 37 | 91 | 29 | 67 |
| TRADES | 114 | 108 | 210 | 307 | 81 | 197 |
| AFD | 276 | 126 | 316 | 285 | 101 | 231 |
| IGR | 9 | 11 | 19 | 25 | 13 | 10 |
| JR | 13 | 47 | 23 | 39 | 14 | 31 |
| **IGSG** | 39 | 117 | 82 | 124 | 71 | 89 |

Table 10: Additional Ablation Study. Adversarial Accuracy and Accuracy over clean testing samples (%) for *IGSG* variants for the MLP model.

| Dataset | Adversary | IGSG-VSG | SGSG | IGIG | $L_2$-IGSG | **IGSG** |
|---------|-----------|----------|------|------|-----------|----------|
| Splice | FSGS | 40.4±3.5 | 41.5±4.1 | 15.6±8.2 | 40.2±1.1 | **44.0±2.6** |
| | OMPGS | 56.3±5.9 | 59.2±8.6 | 45.9±3.5 | 57.9±0.9 | **63.8±4.2** |
| | Clean | 95.7±1.4 | 94.1±0.4 | 90.7±7.9 | 96.0±0.4 | 95.9±0.7 |
| PEDec | FSGS | 85.7±2.2 | 11.9±2.5 | 86.4±2.2 | 81.7±2.6 | **86.5±3.8** |
| | OMPGS | 84.5±3.1 | 30.6±2.1 | 85.7±4.6 | 83.0±1.4 | **88.0±4.0** |
| | Clean | 95.3±0.3 | 96.3±0.1 | 95.3±0.4 | 95.4±0.2 | 95.5±0.2 |
| Census | FSGS | 56.8±3.6 | 66.5±2.1 | 50.2±2.3 | 62.5±1.1 | **67.2±3.5** |
| | OMPGS | 68.6±4.6 | **71.6±6.8** | 62.3±4.2 | 70.6±2.4 | 71.3±9.0 |
| | Clean | 95.3±0.3 | 95.1±0.3 | 95.5±0.1 | 95.3±0.0 | 95.5±0.2 |

to *Adv Train*, *AFD* and *TRADES*. On *Census*, *Fast-BAT*, *JR* and *IGR* need less time than *IGSG*, but there is a large gap between the time cost of *IGSG* and that of those methods.

## I.6 DETAILED ABLATION STUDY

Here, we introduce another three variants of *IGSG*.

*SGSG*: We replace the TV loss of the IG scores with the TV loss defined over the smoothed gradient given in Eq.19.
*IGIG*: Instead of penalizing the $l_p$ norm of the smoothed gradient, we choose to penalize the norm of the IG score vector of each instance $x$. We use *SGSG* and *IGIG* to verify the validity of the two robustness-enhancing regularization terms.
*IGSG-VSG*: We replace the difference of gradient computing given in Eq.5 with the standard smoothed gradient (Smilkov et al., 2017). We introduce *IGSG-VG* and *IGSG-VSG* to demonstrate the necessity of introducing the mean field smoothing-driven smoothed gradient (given by Eq.19) into the gradient smoothing-based regularization term.

$L_2$-*IGSG*: To achieve attribution smoothing, $L_2$ norm regularization is also simple and widely used. We replace the TV loss with an $L_2$ norm of the IG score. We introduce it to further confirm the effectiveness of the TV loss design in *IGSG*.

In Table 10, we provide the adversarial accuracy of the four variants—*IGSG-VSG*, *IGIG*, *SGSG* and $L_2$-*IGSG* —under FSGS attack and OMPGS attack with a budget of 5 for the three datasets on an MLP model. We also compare their performance with that of *IGSG*.

*SGSG* replaces the total variation (TV) loss of *IGSG* with the TV loss of the smoothed gradient. It exhibits slightly inferior performance compared to *IGSG* on the *Splice* and *Census* datasets but performs poorly on the *PEDec* dataset. This can be attributed to the fact that regularizing the TV loss of the smoothed gradient evenly distributes the sensitivity of each feature. However, the gradient information only reflects local sensitivity and does not provide a comprehensive understanding of feature contribution.

*IGIG* replaces the regularization of the smoothed gradient with the $L_Q$ norm of the IG score. Without the use of smoothed sampling, the smoothness of the classifier is inferior to that of *IGSG*. Additionally, IG captures global information about feature contributions but is not as explicit as the gradient in guiding the direction of attack for each category. Therefore, minimizing the magnitude of IG is not as beneficial for the *Splice* and *Census* datasets.

$L_2$-IGSG replaces the TV loss in the regularization of integrated gradient with an $L_2$ norm. Compared to *SG*, $L_2$-IGSG generally has better adversarial accuracy. However, the $L_2$ norm-regulated IG term consistently yields a little lower adversarial accuracy when subjected to FSGS and OMPGS attacks, showing the effectiveness of the TV loss.

In Table 11, we present the accuracy and adversarial accuracy under FSGS attack and OMPGS attack for the Transformer model. The results are similar to those of the MLP model. Compared to the performance of *IGR* shown in Table 7 and Figure 8, *SG* achieves slightly better adversarial robustness due to the smoothing. The only exception is the adversarial accuracy under OMPGS

Table 11: Ablation Study. Adversarial Accuracy and Accuracy over clean testing samples (%) for *IGSG* variants for the Transformer model.

| Dataset | Adversary | SG | IG | IGSG-VG | IGSG-VSG | SGSG | IGIG | **IGSG** |
|---------|-----------|-----|-----|---------|----------|------|------|----------|
| Splice | FSGS | 0.3±0.2 | 2.2±1.6 | 1.5±1.4 | 0.7±0.7 | 1.0±1.0 | 1.3±1.3 | **2.3±1.4** |
| | OMPGS | 33.3±3.7 | 34.9±1.3 | 36.1±4.1 | 34.5±5.0 | 35.9±5.5 | 33.2±3.1 | **36.8±4.3** |
| | Clean | 96.1±0.6 | 96.5±0.5 | 96.7±0.4 | 96.7±0.3 | 96.4±0.6 | 96.7±0.5 | 96.7±0.7 |
| PEDec | FSGS | 60.4±4.4 | 57.1±6.0 | 53.9±3.6 | 60.6±4.3 | 59.9±5.4 | 57.2±6.8 | **63.5±3.7** |
| | OMPGS | **95.7±0.2** | 95.6±0.1 | 95.2±0.3 | 95.2±0.1 | 92.4±2.8 | 91.8±6.6 | 95.6±0.2 |
| | Clean | 95.8±0.3 | 95.7±0.1 | 95.3±0.4 | 95.5±0.2 | 95.1±0.4 | 95.6±0.1 | 95.7±0.3 |
| Census | FSGS | 28.6±0.7 | 31.1±1.1 | 36.6±4.5 | 33.6±2.5 | 28.9±0.9 | 26.2±2.2 | **37.8±4.3** |
| | OMPGS | 56.9±1.1 | 68.7±6.1 | 70.1±6.5 | 73.4±7.2 | 58.3±1.5 | 63.3±2.5 | **76.9±4.8** |
| | Clean | 95.0±0.0 | 94.9±0.1 | 95.0±0.3 | 93.6±0.0 | 95.0±0.0 | 95.2±0.0 | 94.8±0.1 |

Table 12: Adversarial accuracy of IGSG and baseline models on MLP and Transformer model structures under PCAA attack for the three datasets.

| Model | Dataset | Undefended Std Train | Adversarial Training baselines | | | | | Regularization baselines | | Ours |
|-------|---------|--------|----------|----------|--------|--------|--------|---------|--------|--------|
| | | | Adv Train | Fast-BAT | TRADES | AFD | PAdvT | IGR | JR | IGSG |
| MLP | Splice | 37.2±4.0 | 42.6±1.9 | 28.7±7.4 | 27.3±2.4 | 25.8±2.4 | 23.2±4.0 | 42.5±6.0 | 3.5±4.0 | **44.9±2.0** |
| | PEDec | 94.4±0.2 | 94.8±0.2 | 95.6±0.2 | **95.8±0.2** | 94.7±0.2 | 94.9±0.1 | 95.6±0.2 | 95.1±0.2 | 94.7±0.3 |
| | Census | 92.0±0.7 | **93.9±0.1** | 93.1±0.7 | 88.8±0.8 | 93.2±0.1 | 93.4±0.4 | 93.6±0.0 | 93.4±0.1 | 93.8±0.0 |
| Transformer | Splice | 8.6±3.9 | 2.8±0.9 | 10.5±3.2 | 7.3±1.2 | 2.4±1.7 | 6.5±1.8 | **11.3±3.5** | 7.8±3.4 | 11.1±2.6 |
| | PEDec | 87.1±2.4 | 75.5±1.2 | 87.8±0.9 | **90.8±0.6** | 86.7±3.3 | 87.4±1.2 | 89.7±2.3 | 90.6±1.0 | 89.2±1.3 |
| | Census | 92.3±1.0 | 94.5±0.3 | 91.8±1.2 | 91.5±1.9 | 92.9±1.2 | **94.3±0.2** | 93.3±0.3 | 93.8±0.7 | 93.7±0.2 |

attack for *PEDec*, where *SG* achieves much better robustness. This may be a result of the smoothness of gradients among neighboring samples. Notably, most variants of *IGSG* achieve very high adversarial accuracy under OMPGS attack, suggesting that both *IG* and *SG* training can defend against OMPGS attack on *PEDec*. Regarding *IG*, *IGSG-VG*, and *IGSG-VSG*, their performance varies across datasets, indicating instability. On the other hand, *SGSG* and *IGIG* do not perform well on any dataset, suggesting that the roles of IG and SG cannot be effectively altered by each other in the loss function.

## I.7  ROBUSTNESS EVALUATION UNDER PROBABILISTIC CATEGORICAL ADVERSARIAL ATTACK (PCAA)

In this section, we assess the robustness of our proposed IGSG and baseline methods against the PCAA attack (Xu et al., 2023) on three datasets using MLP and Transformer models. The evaluation maintains a consistent setting from previous experiments, with a budget limit of 5 for each dataset.

Table 12 presents the outcomes of this evaluation. It is evident that PCAA does not ensure uniform effectiveness across different datasets. When compared with the results in Table 2 and Table 7, PCAA demonstrates comparable effectiveness to FSGS in attacking the *Splice* dataset with the MLP model and slightly lesser efficacy with the Transformer model. However, its performance on the *PEDec* and *Census* datasets is markedly weaker. The adversarial accuracy for undefended models remains above 87% for both model architectures on these datasets. This could be due to *PEDec*'s high-dimensional feature space and the diverse and extensive categorical dimensions in *Census*, suggesting that PCAA is not an effective measure for assessing robustness in these contexts.

In terms of adversarial accuracy under PCAA attack, IGSG excels on the *Splice* dataset with the MLP model and attains second-best performance with the Transformer model, closely trailing IGR. Although IGSG does not show high adversarial accuracy on the *PEDec* and *Census* datasets compared to the baselines under PCAA attack, this attack strategy is not a reliable measure for these datasets due to its limited effectiveness. Nonetheless, the results from the *Splice* dataset indicate that IGSG notably enhances model robustness.