# OpenReview forum: "Enhancing Adversarial Robustness on Categorical Data via Attribution Smoothing"
_ICLR.cc/2024/Conference — Submitted to ICLR 2024_

### Official Review · Reviewer_rFyD · 2023-10-30

**Soundness:** 3 good
**Presentation:** 3 good
**Contribution:** 3 good
**Rating:** 6
**Confidence:** 3

**Summary:**

This paper proposes a regularization method to enhance the robustness of classification over categorical attributes against adversarial perturbations. The paper establishes an information-theoretic upper bound on the expected adversarial risk and proposes an adversarially robust learning method, named Integrated Gradient-Smoothed Gradient (IGSG)-based regularization, which designed to smooth the attributional sensitivity of each feature and the decision boundary of the classifier to achieve lower adversarial risk, desensitizing the categorical attributes in the classifier. The paper conducts extensive empirical study over categorical datasets of various application domains to confirm the effectiveness of IGSG and achieve new start-of-arts.

**Strengths:**

This paper has good originality, high quality and clear expression. The paper proposes a new regularization method which outperforms adversarial training and generalize well.

**Weaknesses:**

Larger dataset is better to be verified to demonstrate the genelarization of the proposed method.

**Questions:**

1.This paper argues that the proposed method can smooth the decision boundary of the classifier, how about to visualize the decision boundary?
2.Is there exists obfuscated gradients in the proposed method?
3.Is the proposed method also works well under other attacks such as deepfool attack, C&W attack?

---

> ### Author Response · Authors · 2023-11-19
> **Visualizations of the classification boundary**
>
> Many thanks to the viewer for the insightful questions. In the response below, we first provide answers to these questions, and then would like to discuss more about the changes we made in the updated draft, thanks to the insightful questions from the reviewer.
>
> **Visualizations of the classification boundary**: Thank you for your advice, we have visualized and compared the decision boundaries of classifiers trained using IGSG and Std Train, specifically focusing on the PEDec dataset. The results of this analysis can be found in Appendix I.2. Compared to Std Train, IGSG facilitates a smoother and more discernible classification boundary. Such a visualization not only highlights the distinctiveness of the IGSG method but also demonstrates its efficacy in achieving clearer class separations.

---

> ### Author Response · Authors · 2023-11-19
> **Discussions about the obfuscated gradients**
>
> **Discussions about the obfuscated gradients**:
> Gradient obfuscation is not a concern with IGSG, as evidenced by several factors. Firstly, IGSG maintains access to all gradients, effectively preventing the occurrence of shattered gradients. Secondly, the absence of randomized components in IGSG rules out the possibility of stochastic gradients. Lastly, IGSG's use of smoothed gradient regularization, while potentially diminishing gradient norms, does not undermine its true robustness. This is evident in its high adversarial accuracy against two attack methods, OMPGS and FSGS, which are not influenced by gradient magnitudes. OMPGS focuses exclusively on the gradient order of each feature, and FSGS, being a black-box attack, is impervious to gradient dynamics. The effectiveness of IGSG against these attacks, without relying on reduced gradient norms, underscores its genuine enhancement of model robustness. Consequently, IGSG does not exhibit the three characteristic factors of gradient obfuscation as identified in \cite{Athalye et.al., 2018}, thereby affirming its immunity to gradient obfuscation concerns.
>
> [Athalye et.al., 2018] Athalye A, Carlini N, Wagner D. Obfuscated gradients give a false sense of security: Circumventing defenses to adversarial examples[C]//International conference on machine learning. PMLR, 2018: 274-283.

---

> ### Author Response · Authors · 2023-11-19
> **IGSG performance on other attack methods**
>
> **IGSG performance on other attack methods**: It is worth noting that the use of most existing attack methods, which typically focus on $L_Q$ norm-constrained attacks (where $q \geq 1$), is not feasible in the context of discrete data. This infeasibility arises because adversarial samples produced by such methods do not align with the discrete data space, rendering them infeasible for robustness evaluation in this domain. The incompatibility of $L_Q$ norm-constrained attacks with discrete data spaces is comprehensively illustrated in Appendix G.1. Among these methods, Deepfool and the C&W attack are notable examples. Deepfool employs a gradient method to identify adversarial samples, and it is not tailored for discrete data at all. For the C&W attack, although it introduces an $L_0$ norm-constrained approach, effectively limits only the number of pixels altered, it still generates adversarial samples that, in one-hot encoded form, contain float values. For instance, in the case of a sample $x_i$ with one-hot encoding $b(x_i)$, the C&W attack modifies features $j’ \in \omega_i$ using an $L_2$ attack, leading to non-discrete values in $b(\hat x_i)_{j’}$. Consequently, the adversarial samples generated by the C&W attack also do not reside in the discrete data space, further underscoring the infeasibility of using these methods to evaluate robustness in categorical data contexts.
>
> To address the limitations in our previous experimental setup, we conducted an evaluation of IGSG and other baseline models under the Probabilistic Categorical Adversarial Attack (PCAA) \cite{xu2023probabilistic}, specifically designed for categorical data. This additional analysis is detailed in Appendix I.2. However, it was observed that PCAA does not consistently lead to successful attacks across different datasets. As indicated by the results in Table 12, PCAA's effectiveness is comparable to that of FSGS on the Splice dataset, where IGSG demonstrates relatively superior performance against baseline methods. Nevertheless, on the other two datasets, PCAA fails to effectively attack even the undefended models, rendering it an inefficient tool for evaluating robustness in these cases.
>
> [xu2023probabilistic] Xu H, He P, Ren J, et al. Probabilistic categorical adversarial attack and adversarial training[C]//International Conference on Machine Learning. PMLR, 2023: 38428-38442.

---

> > ### Comment · Reviewer_rFyD · 2023-11-22
> >
> > Thanks to the authors for their reply,I have no more questions.

---

> > > ### Author Response · Authors · 2023-11-22
> > >
> > > Dear Reviewer rFyD,
> > >
> > > Thank you for your response and for the time you have invested in reviewing our work. We appreciate your insightful suggestions and are glad to have the opportunity to address your questions. Please let us know if there is anything else we can provide to assist in this process.
> > >
> > > Best regards,
> > >
> > > The Authors of Submission 4530

---

### Official Review · Reviewer_Bx5T · 2023-10-31

**Soundness:** 3 good
**Presentation:** 3 good
**Contribution:** 3 good
**Rating:** 6
**Confidence:** 2

**Summary:**

This paper proposes IGSG, a theoretically motivated robust learning method for categorical inputs. It contains two parts: Integrated Gradient (IG) and Gradient Regularization. Experiments verify the effectiveness of the proposed method.

**Strengths:**

- The studied problem is interesting, adversarial attack/defense on caterical data is of interest to the community;
- The experimental results are strong to present the effectiveness of the proposed IGSG method.
- The paper is well-written and well formatted.

**Weaknesses:**

- Minor issue:
     - Page 9, "”robust overfitting”"

**Questions:**

- I find some Adversarial Training baseline and Regularization baseline methods perform worse than undefended method. Could you please explain the reason?
- Could IGSG be used or modified to be used in continuous data?
- Could you please explain more about Figure 3? Why the summing of the attack frequency of IGSG features are lower than that of original features?
-  What is the meaning of the sign of $IG$? If the sign of insignificant, maybe using the absolute of $IG$ is better.
- According to Eq.3 and Factor 1, IGSG should base on the adversarial training. However, S is the original dataset used in Eq.5. Why not use the adversarial training as a base?
- I notice a recently ICML 2023 paper *Probabilistic Categorical Adversarial Attack and Adversarial Training* proposes a gradient-based attack. However, the attack methods you used (i.e. FSGS and OMPGS) are both search-based attack method. Therefore, the improvements against other methods your presented might come from the relatively weak attack, because the adversarial training employed gradient-based attack instead of search-based attack. Could you please show more results on defending the gradient-based attack methods?

---

> ### Author Response · Authors · 2023-11-19
> **Explanations of some Adversarial Training baseline and Regularization baseline methods performing worse than the undefended method.**
>
> Many thanks to the viewer for the insightful questions. In the response below, we first provide answers to these questions, and then would like to discuss more about the changes we made in the updated draft, thanks to the insightful questions from the reviewer.
>
> **Explanations of some Adversarial Training baseline and Regularization baseline methods performing worse than the undefended method.**: This is because some baseline adversarial training and regularization methods under consideration are not inherently suited for categorical data. As delineated in Table 1, PGD-based adversarial training can effectively counter the adversarial samples crafted via the PGD-1 attack. However, this approach is less effective against adversarial samples from attacks tailored for discrete data (FSGS and OMPGS). FSGS and OMPGS produce adversarial examples with a significantly different distribution compared to those generated by gradient-based attacks like PGD-1. All the adversarial training-based methods suffer the same problem, leading to inconsistent performance across various datasets. Regarding regularization-based methods, Jacobian Regularization (JR) occasionally underperforms compared to Std Train. This could be attributed to JR's approach of regularizing the gradient of each class's logits with respect to input features. This process excessively penalizes boundary smoothness, which can sometimes reduce adversarial robustness.

---

> ### Author Response · Authors · 2023-11-19
> **Using IGSG in continuous data**
>
> **Using IGSG in continuous data**: In the application of IGSG, tabular data of various types can be effectively utilized. As an illustrative case, we consider the Census dataset, selected for its composition of both categorical and numerical variables. Notably, the numerical variables in this dataset represent continuous data, providing a comprehensive example for demonstrating the versatility of IGSG in handling diverse data formats.

---

> ### Author Response · Authors · 2023-11-19
> **Explanations of Figure 3**
>
> **Explanations of Figure 3**: Figure 3 illustrates the frequency of attacks on individual features across the entire test dataset. For a given data point $x_i$, if a successful attack is executed by perturbing features $j_1, j_2, \ldots, j_s$, the attack frequency for each of these $s$ features is incremented by one. This process is repeated for the entire dataset, culminating in the results displayed in Figure 3. The observed lower attack frequencies for features in the model trained using IGSG, compared to those in the Standard Training (Std Train) model, indicate that Std Train possesses certain features that are more vulnerable to adversarial attacks. In contrast, IGSG training appears to mitigate this vulnerability.

---

> ### Author Response · Authors · 2023-11-19
> **Meaning of the signs of IG scores**
>
> **Meaning of the signs of IG scores**: When the Integrated Gradients (IG) score for a feature is negative, it suggests that the feature negatively influences the classification of the sample towards the ground truth label. Conversely, a positive IG score indicates that the feature positively influences this classification. Therefore, the sign of the IG score is of significant importance, as it provides insight into the influence of each feature on the model's decision-making process.

---

> ### Author Response · Authors · 2023-11-19
> **Reasons for not using adversarial training as a base for IGSG**
>
> **Reasons for not using adversarial training as a base for IGSG**: We are sorry for the ambiguous saying. We demonstrate the limitations of adversarial training for categorical data in Section 3.1. The primary challenge with adversarial training in this context is its inability to generate a comprehensive array of adversarial examples that encompass the entire discrete space, leading to a performance deficit.
>
> This limitation is empirically substantiated in Table 1, where a significant distribution shift is evident between adversarial samples generated through gradient-based methods, like PGD-1 attack, and those derived from greedy search-based methods, like FSGS and OMPGS. The disparity underscores that gradient-based approaches do not capture a substantial segment of the adversarial example space. Consequently, numerous adversarial scenarios remain unexplored and unaccounted for by these methods.
>
>  In response to this challenge, our approach strategically diverges from relying on adversarial training as the foundational technique. Instead, we employ regularization-based methods for our algorithmic design.

---

> ### Author Response · Authors · 2023-11-19
> **Augmented experiments on baseline PAdvT and attack method PCAA, and the effectiveness of greedy search-based attack methods for categorical data**
>
> **Augmented experiments on baseline PAdvT and attack method PCAA, and the effectiveness of greedy search-based attack methods for categorical data**: Thank you for the advice, we have conducted additional experiments employing PCAA as an attack method to assess the adversarial robustness of various models in Appendix I.7. We also include a comparison between IGSG and PAdvT, where PAdvT utilizes adversarial training with PCAA-generated adversarial samples for discrete data. The results for PAdvT are in Table 2, Figure 2. Our findings indicate that PAdvT, as proposed in PCAA, generally underperforms relative to IGSG, and in many instances, is even less effective than AdvTrain. It could be attributed to its methodology. Unlike AdvTrain, which applies PGD adversarial training directly to smoothed categorical data, PAdvT first samples adversarial examples from PGD-generated samples within the discrete space and then proceeds with adversarial training on these samples. This results in a distribution shift from the optimal inner maximization achievable by gradient-based methods, leading to less effective training.
>
> Furthermore, we underscore the appropriateness of greedy search-based attack methods for handling categorical data. Given that adversarial attacks on discrete data constitute a combinatorial optimization problem, greedy search-based methods such as FSGS are theoretically sound and have proven to be highly effective within computationally feasible boundaries. Although OMPGS, which combines greedy search with gradient-based filtering, exhibits slightly reduced effectiveness compared to FSGS, it significantly improves efficiency.
>
> Finally, our inclusion of the PCAA attack in the comparison further reinforces our findings. In the majority of scenarios, PCAA struggles to execute effective attacks, thus proving inadequate for assessing the robustness of both our proposed methods and other baseline models.

---

### Official Review · Reviewer_KxJ6 · 2023-10-31

**Soundness:** 2 fair
**Presentation:** 2 fair
**Contribution:** 2 fair
**Rating:** 5
**Confidence:** 4

**Summary:**

This work studies adversarial robustness on tabular categorical attributes and establishes an information-theoretic upper bound on the expected adversarial risk.

**Strengths:**

- Provide some theoretical results.
- Easy to follow.

**Weaknesses:**

- This work develops the theory over randomized learning algorithm $A(\cdot)$, however, the experiments are deployed on deterministic models. How can these empirical results support the theory?
- Given $S$, where does the randomness of $A(S)$ come from? Could authors provide a simple case of randomized algorithm that the theory can totally hold?
- I am familiar with PAC Bayes, which involves a posterior distribution of $A(S)$. Does a posterior distribution of $A(S)$ also exist here? If so, what assumptions are made about $A(S)$? If not, could authors provide the precise definition of $A(S)$?
- In $I(f,S)$, $f$ and $S$ are random variables, how to use $IG(x)$ with a given deterministic $f$ to influence $I(f,S)$ with a random $f$. We even have no idea with the assumption of random $f$. It is also confused which $f$ is random and which $f$ is deterministic, they are both represented as $f$.
- The adversarial robustness of tabular categorical attributes is only a minor extension of DNNs adversarial robustness, the contribution seems minor.

**Questions:**

See weaknesses.

---

> ### Author Response · Authors · 2023-11-19
> **How does the assumption of randomized learning apply to the experiments?**
>
> Many thanks to the viewer for the insightful questions. In the response below, we first provide answers to these questions, and then would like to discuss more about the changes we made in the updated draft, thanks to the insightful questions from the reviewer.
>
> **How does the assumption of randomized learning apply to the experiments?**: To avoid possible confusion, we have revised Section 3.2 and Appendix D to provide a clarification of the rationale behind the randomized learning assumption. Our investigation is grounded in the exploration of mutual-information-based generalization error bounds, as outlined in the work of Xu et al. \cite{Xu2017NiPS}. In this line of research, a machine learning algorithm is conceptualized as a randomized mapping or an information-transmitting channel, employing the language of information theory. This mapping or channel takes a training dataset as input and yields a hypothesis as output. The randomness inherent in this mapping/channel manifests in two dimensions. First, the training dataset provided to the channel is a sample selected from all possible combinations of $n$ training data points. Second, the resulting hypothesis from this channel is one sample chosen from all possible hypotheses within the hypothesis space. The mutual-information-based bound in Eq.3 then provides an upper bound to the expected adversarial risk over all possible hypothesis functions. In other words, we offer an averaged estimate of the potential adversarial risk, irrespective of the hypothesis chosen as the output by the learning algorithm. More concretely, in the context of a concrete learning task, our mutual-information-based bound is applicable to a classifier, irrespective of whether the parameters or decision outputs of this classifier are deterministic or randomized.
>
> In Section 3.2, we refine the definition of the expected and empirical adversarial risk in Definition.1, as well as the formulation of the adversarial risk upper bound in Theorem.1. We highlight that both expected and empirical adversarial risk are derived by averaging over all possible hypothesis functions and possible training dataset $S^n$. This is consistent with the fundamental setting in the study of the mutual-information-based generalization error bound.
>
> [Xu2017NiPS] Aolin Xu and Maxim Raginsky. 2017. Information-theoretic analysis of generalization capability of learning algorithms. In Proceedings of the 31st International Conference on Neural Information Processing Systems (NIPS'17). Curran Associates Inc., Red Hook, NY, USA, 2521–2530.

---

> ### Author Response · Authors · 2023-11-19
> **A simple case of randomized algorithm that the theory can totally hold**
>
> **A simple case of randomized algorithm that the theory can totally hold**: Two instances of randomized learning mechanisms could be 1) Gibbs algorithm for optimization raised in \cite{Xu2017NiPS} and 2) Stochastic Gradient Langevin Dynamics used \cite{Bu et.al., 2020}. \cite{Xu2017NiPS, Bu et.al., 2020} provides the mutual-information-based generalization error bound for the learning algorithm using Gibbs optimization and SGLD respectively.
>
> [Xu2017NiPS] Aolin Xu and Maxim Raginsky. 2017. Information-theoretic analysis of generalization capability of learning algorithms. In Proceedings of the 31st International Conference on Neural Information Processing Systems (NIPS'17). Curran Associates Inc., Red Hook, NY, USA, 2521–2530.
>
> [Bu et.al., 2020] Bu Y, Zou S, Veeravalli V V. Tightening mutual information-based bounds on generalization error[J]. IEEE Journal on Selected Areas in Information Theory, 2020, 1(1): 121-130.

---

> ### Author Response · Authors · 2023-11-19
> **What assumptions are made about the randomized learning mechanism?**
>
> **What assumptions are made about the randomized learning mechanism?**: Following \citep{Xu2017NiPS, Bu et.al., 2020}, we don't impose any prior distribution assumption over the learning mechanism $P_{f|S^n}$ or the hypothesis space. This characterizes the major difference between our study and PAC-Bayes generalization bounds \citep{McAllester99PACBayes}. The mutual information-based analysis involves determining a bound to the generalization error that holds on average. For example, \cite{Xu2017NiPS} derived average generalization error bounds involving the mutual information between the training set and the hypothesis. Though PAC-Bayesian bounds also connect information-theoretic quantities to generalization and are similar to the mutual information approach, these bounds are output-dependent, PAC-Bayesian bounds provide a generalization bound for a particular output hypothesis or hypothesis distribution, rather than uniformly bounding the expected error of the algorithm as obtained in the mutual-information based bound (Eq.3). For example, the original PAC-Bayesian generalization error bounds is characterized via a Kullback-Leibler (KL) divergence (a.k.a. relative entropy) between a prior data-free distribution and a posterior data dependent distribution on the hypothesis space. We also update this discussion to Appendix C.
>
> [Xu2017NiPS] Aolin Xu and Maxim Raginsky. 2017. Information-theoretic analysis of generalization capability of learning algorithms. In Proceedings of the 31st International Conference on Neural Information Processing Systems (NIPS'17). Curran Associates Inc., Red Hook, NY, USA, 2521–2530.
>
> [Bu et.al., 2020] Bu Y, Zou S, Veeravalli V V. Tightening mutual information-based bounds on generalization error[J]. IEEE Journal on Selected Areas in Information Theory, 2020, 1(1): 121-130.
>
> [McAllester99PACBayes] McAllester D A. Some pac-bayesian theorems[C]//Proceedings of the eleventh annual conference on Computational learning theory. 1998: 230-234.

---

> ### Author Response · Authors · 2023-11-19
> **The adversarial robustness of tabular categorical attributes is only a minor extension of DNNs adversarial robustness**
>
> **The adversarial robustness of tabular categorical attributes is only a minor extension of DNNs adversarial robustness**: We respectfully disagree with this comment. Our core contribution is to unveil the intrinsic difference between organizing adversarial defense on categorical data and that on continuous data. This difference causes the bottleneck of applying the well-known adversarial training-based defense strategy to the categorical data.
>
> While adversarial defense on continuous data, such as pixels, can be effectively conducted as an adversarial training process,  our work reveals that the nature of adversarial defense on high-dimensional categorical data fundamentally constitutes a challenging Mixed Integer Nonlinear Programming (MINLP) problem. In this problem, generating adversarial samples is a combinatorial search process. Gradient-guided methods, e.g. PGD/FGSM attack, are no longer applicable. The exponential growth of the categorical adversarial space with increasing amounts of categorical features further complicates the generation of adversarial samples.
>
> Empirical findings presented in Section 3.1 demonstrate that the challenge of navigating the categorical adversarial space contributes to an insufficient exploration of feasible adversarial samples during adversarial training. Consequently, this inadequacy results in a distribution gap between the adversarial training samples and those utilized in subsequent attacks. Ultimately, this discrepancy leads to the robust overfitting of defense strategies based on adversarial training when applied to categorical data. Our comparative analysis in Section 5.2 serves to 1) validate the ineffectiveness of adversarial training-based methods in enhancing adversarial robustness on categorical data and 2) closely align with the theoretical analysis based on mutual information concerning the adversarial risk bound on categorical data.

---

### Official Review · Reviewer_iddk · 2023-11-01

**Soundness:** 3 good
**Presentation:** 3 good
**Contribution:** 3 good
**Rating:** 6
**Confidence:** 4

**Summary:**

The authors address adversarial robustness in the context of categorical variables, a less-explored scenario in the literature. They highlight the key finding that different attack methods yield adversarial samples with varied distributions. Relying solely on PGD-generated adversarial samples for training leads to overfitting and inadequate defense against diverse attacks. Consequently, they opt to bolster robustness in categorical data training through a regularization-based approach.

By establishing an upper bound on the theoretical risk gap and analyzing factors that mitigate adversarial risk, they propose IGSG—a regularization-based paradigm for robust training with categorical variables. Experimental results across multiple datasets affirm the method's superiority over baselines and traditional adversarial training techniques.

In summary, the authors introduce IGSG, a regularization-based robust training method, grounded in a theoretical analysis of factors reducing adversarial risk in dealing with categorical variables. Empirical evidence demonstrates its outperformance over baselines and competing adversarial training methods.

**Strengths:**

1、The authors address adversarial robustness in the context of categorical variables, an aspect that has received limited exploration in the literature.
2、The authors present a clear motivation for their work.
3、The authors substantiate their motivation through explicit theoretical and empirical experiments.

**Weaknesses:**

1、The expression contains ambiguities, such as the specific definition of $G^r" in equation (5), which needs clarification in the main text. On page six, the authors mention minimizing the "third and fourth terms" in the first line, but the subsequent explanations actually refer to optimizing the second and third terms. For the cross-validation of hyperparameters, it would be beneficial for the authors to elaborate on how they performed the training-validation set split using only the training data and specify the chosen values for the hyperparameters.
2、The experiments appear insufficient, and I suggest the authors supplement the following experiments to bolster support for their method:
1）While MLP and Transformer serve as baseline models, including more models would validate the generalizability of their method.
2）Despite utilizing cross-validation for hyperparameter selection, reporting performance with different hyperparameter choices is recommended to assess the sensitivity of their method.

**Questions:**

None

---

> ### Author Response · Authors · 2023-11-19
> **Ambiguities and the settings of cross-validation of hyperparameters**
>
> Many thanks to the viewer for the insightful questions. In the response below, we first provide answers to these questions, and then would like to discuss more about the changes we made in the updated draft, thanks to the insightful questions from the reviewer.
>
> **Ambiguities and the settings of cross-validation of hyperparameters**: Thank you for pointing out the ambiguities. The revised manuscript has effectively addressed the noted ambiguities. Notably, enhancements include a refined version of Eq.5. Also, detailed information on hyperparameters is stated at the end of the second paragraph in Appendix H.3. Moreover, the latest version incorporates an elaborate discussion on an explanation of the setting of cross-validation to hyperparameters, situated in the same paragraph of Appendix H.3, highlighted with the blue color font. We hope that this updated information can help readers better comprehend the equations and hyperparameter settings described in our paper.

---

> ### Author Response · Authors · 2023-11-19
> **Augmented experiments**
>
> **Augmented experiments**: Thanks for the inspiring suggestions. In response to the concerns about model augmentation, CNN-based model architectures use a sliding window when processing the original data, and RNN-based model architectures are designed for sequential data. However, these two models may not be ideally suited for datasets like PEDec and Census where sequential or neighboring feature relationships are absent. This justifies our choice of MLP and Transformer models, which do not rely on positional information. According to the limitations in computational resources, we regret the inability to perform an exhaustive comparison across various hyperparameter settings.
>
> We also realize that the experiment appears insufficient. Thus, we add one more baseline model PAdvT \cite{xu2023probabilistic} to give a more comprehensive comparison with IGSG and evaluate IGSG and all the baseline models under a new attack method PCAA \cite{xu2023probabilistic}. The relevant results and updates are detailed in Table 2, Figure 2, Table 7, Figure 8, and Appendix I.7. The findings consistently show IGSG's superiority over PAdvT across all three datasets and its effective defense against PCAA attacks. Furthermore, to illustrate IGSG's impact in smoothing the classification boundary, a visualization of the classification boundary for the PEDec dataset is provided in Appendix I.2. Lastly, to empirically support the theoretical assertions, a toy model approximation of mutual information in Eq.3 is presented in Appendix I.1, using the Splice and PEDec datasets. This experiment demonstrates that IGSG indeed lowers the mutual information bound as stated in Eq.3, correlating with a reduced adversarial risk bound and consequently leading to enhanced adversarial accuracy.
>
> [xu2023probabilistic] Xu H, He P, Ren J, et al. Probabilistic categorical adversarial attack and adversarial training[C]//International Conference on Machine Learning. PMLR, 2023: 38428-38442.

---

> > ### Author Response · Authors · 2023-11-23
> > **Test Across Various Hyperparameter Settings**
> >
> > **Testing Across Various Hyperparameter Settings**: We appreciate the recommendations to test under different hyperparameter configurations to evaluate sensitivity. Given the extensive range of combinations, we have opted to fix one parameter at a time while examining the sensitivity of another. The following section presents our findings with varying $\alpha$ and varying $\beta$ respectively:
> >
> > | Dataset | Settings   | Attack | α=0.01 | α=0.1 | α=1   | α=10  |
> > |---------|------------|--------|--------|-------|-------|-------|
> > | Splice  | β=100      | FSGS   | 48.2   | 43.5  | 41.1  | 44.2  |
> > |         |            | OMPGS  | 65.2   | 61.7  | 66.1  | 58.9  |
> > | PEDec   | β=0.1      | FSGS   | 85.8   | 75.2  | 91.1  | 77.4  |
> > |         |            | OMPGS  | 90.4   | 90.6  | 94.8  | 91.4  |
> > | Census  | β=3.0      | FSGS   | 62.0   | 59.3  | 66.7  | 70.7  |
> > |         |            | OMPGS  | 67.5   | 85.9  | 72.4  | 84.5  |
> >
> > | Dataset | Settings   | Attack | β=0.01 | β=0.1 | β=1   | β=10  |
> > |---------|------------|--------|--------|-------|-------|-------|
> > | Splice  | α=10       | FSGS   | 40.7   | 39.9  | 42.6  | 43.4  |
> > |         |            | OMPGS  | 56.0   | 59.2  | 56.1  | 60.2  |
> > | PEDec   | α=0.01     | FSGS   | 76.2   | 78.3  | 84.2  | 79.2  |
> > |         |            | OMPGS  | 93.6   | 93.4  | 94.4  | 94.6  |
> > | Census  | α=1        | FSGS   | 57.7   | 72.9  | 58.5  | 63.0  |
> > |         |            | OMPGS  | 69.5   | 80.2  | 73.5  | 90.7  |
> >
> >
> > Given the time constraints, we conducted testing on a single experiment. The results indicate that variations in hyperparameter settings do not substantially affect IGSG's performance. Typically, achieving a balance among the three losses outlined in Equation 5 suffices to obtain an effective solution.

---

### Official Review · Reviewer_eEhS · 2023-11-01

**Soundness:** 2 fair
**Presentation:** 3 good
**Contribution:** 2 fair
**Rating:** 5
**Confidence:** 4

**Summary:**

This paper studies adversarial robustness of classification over categorical attributes against adversarial perturbation. An information-theoretic upper bound on the expected adversarial risk has been established, together with an adversarial learning method with integrated gradient-smoothed gradient regularization. Experimental results demonstrate the effectiveness of the proposed training method and the superiority to the state-of-the-art robust training methods in terms of adversarial accuracy.

**Strengths:**

This paper is well-motivated and well-written. The adversarial robustness on categorical data is an interesting topic, and this paper takes a reasonable approach to tackle related challenges. Some interesting insights are also discussed, and they would be of interest to the community.

The definition of adversarial risk for categorical data is reasonable. It appears the information theoretic upper bound on the expected adversarial risk is new, and the remarks on the upper bound are sensible. The regularization terms inspired from the bound are reasonable.

There are some interesting experiments in the appendix, which are good to know.

**Weaknesses:**

It seems there is some gap between the information-theoretic upper bound and the regularization terms. They are weakly linked by some factors implied by the upper bound, but there does not seem there are any direct connection between them. That is, the factors are quite intuitive in such a way that the regularization terms can be designed without knowing the information-theoretic bounds. The mutual information terms just state some weak dependence between random variables, but there is no mention on how to compute/approximate them in the current context. It is unclear how tight the upper bound could be, and the evaluation of the dependence of the bounds against key parameters in practical models and datasets is also missing.

The proposed training method just puts two regularization term together with some hyper-parameters without explaining why they should be composed in that way. It is unclear how these regularization terms are connected to the mutual information terms in the upper bound.

**Questions:**

1.	State clearly and explicitly how the information-theoretic upper bound is connected to the regularization terms.
2.	Evaluate the upper bound against the key parameters for practical models and datasets.

---

> ### Author Response · Authors · 2023-11-19
> **Disconnection between the theoretical analysis and the design of the algorithm.**
>
> Many thanks to the viewer for the insightful questions. In the response below, we first provide answers to these questions, and then would like to discuss more about the changes we made in the updated draft, thanks to the insightful questions from the reviewer.
>
> **Disconnection between the theoretical analysis and the design of the algorithm.**: Thanks for the comments. We revise Section 3.2, Section 4 and Appendix B to clarify how the algorithmic design is rooted in the information-theoretical analysis of the adversarial risk bound. This revision aims to provide a clearer understanding of how the mutual information factors directly relate to and influence the effectiveness of the proposed regularization strategy.
>
> Our logic of reasoning can be summarized into two steps:
>
> **First Argument Step: analysis of two factors suppressing the adversarial risk**: In Section 3 and Appendix B, we illustrate that the information-theoretical bound in Eq.3 unveils two major factors to suppress the adversarial risk over categorical inputs.
>
> Factor 1. Reducing $I(f;z_{i})$ for each training sample $z_i$ helps suppress the adversarial risk of $f$. $I(f, z_i)$ in Eq.3 represents the mutual information between the classifier $f$ and each training sample $z_i$. Pioneering works \cite{Xu2017NiPS,Bu et.al., 2020,Zhang21tnnls} have established that a lower value of $I(f, z_i)$ corresponds to a diminishing adversary-free generalization error. As widely acknowledged in adversarial learning research and emphasized in Eq.3, a better generalizable classifier exhibits greater resilience to adversarial attacks, resulting in lower adversarial risk.
>
> Factor.2 Reducing $\Psi(x_{i,\omega_i},x_{i,\overline{\omega_i}})$ and $\Phi(x_{i,\omega_i},\hat{x}_{i,{\omega_i}})$ helps smooth the feature-wise contribution to classification, thus reducing the adversarial risk.
>
> First, in $\Psi(x_{i,\omega_{i}},x_{i,\overline{\omega_i}})$, $I(x_{i,\omega_i};f)$ and $I(x_{i,\overline{\omega_i}},y_i;f)$ reflect the contribution of the feature subset $\omega_i$ and the rest features $\overline{\omega_i}$ to $f$.
> Features with higher mutual information have a more substantial influence on the decision output, i.e. adversarially perturbing the values of these features is more likely to mislead the decision. Minimizing $\Psi(x_{i,\omega_i},x_{i,\overline{\omega_i}})$ thus decreases the contribution gap between the attacked and untouched features. It prompts the classifier to maintain a more balanced reliance on different features, thereby making it harder for adversaries to exploit influential features.
>
> Second, $\Phi(x_{i,\omega_i},\hat x_{i,{\omega_i}})$ measures the sensitivity of features in $\omega_i$, in terms of how adversarial perturbations over this subset of features affect both the classification output and the correlation between $\omega_i$ and $\overline{\omega_i}$. Minimizing $\Phi(x_{i,\omega_i},\hat x_{i,{\omega_i}})$ makes the classifier's output less sensitive to the perturbations over input features, which limits the negative impact of adversarial attacks. In conclusion, jointly minimizing $\Psi(x_{i,\omega_i},x_{i,\overline{\omega_i}})$ and $\Phi(x_{i,\omega_i},\hat x_{i,{\omega_i}})$ ensures that the classifier does not overly rely on a few highly sensitive features. It helps reduce the susceptibility of the classifier to adversarial perturbation targeting these features, which consequently limits the adversarial risk.
>
> [Xu2017NiPS] Aolin Xu and Maxim Raginsky. 2017. Information-theoretic analysis of generalization capability of learning algorithms. In Proceedings of the 31st International Conference on Neural Information Processing Systems (NIPS'17). Curran Associates Inc., Red Hook, NY, USA, 2521–2530.
>
> [Bu et.al., 2020] Bu Y, Zou S, Veeravalli V V. Tightening mutual information-based bounds on generalization error[J]. IEEE Journal on Selected Areas in Information Theory, 2020, 1(1): 121-130.
>
> [Zhang21tnnls] Zhang J, Liu T, Tao D. An optimal transport analysis on generalization in deep learning[J]. IEEE Transactions on Neural Networks and Learning Systems, 2021.

---

> > ### Author Response · Authors · 2023-11-19
> > **Disconnection between the theoretical analysis and the design of the algorithm. Second Argument Step: how IGSG accords with the two factors mentioned above**
> >
> > **Disconnection between the theoretical analysis and the design of the algorithm.**
> >
> > **Second Argument Step: how IGSG accords with the two factors mentioned above**: In Section 4, we explain how the proposed IGSG method is designed in accordance with the two recommended factors, in order to minimize the adversarial risk. It is worth noting that deriving accurate and differentiable estimation of mutual information between high-dimensional variables, such as the parameters of deep neural networks and input categorical feature vectors, remains an open and challenging problem. This makes direct optimization of the mutual information-based bound impractical. To reach this goal, the proposed IGSG jointly applies two smoothness-enhancing regularization techniques into the learning process of a classifier with categorical inputs, in order to mitigate the adversarial attack over categorical data. The relation between the algorithmic design and the two recommended factors is explained below:
> >
> > **Minimizing $I(f;z_i)$ by smoothing the curvature of the classification boundary.** In previous work, Fisher information $\rho(z_i)_f$ was utilized as a quantitative measure of the information that the hypothesis $f$ contains about the training sample $z_i$ \cite{Hannun2021MeasuringDL}. As shown in \cite{wei2016mutual}, $\rho(z_i)_f$ is closely related to the mutual information $I(f;z_i)$, higher/lower $\rho(z_i)_f$ indicates higher/lower $I(f;z_i)$ . Our proposal aims to minimize $\rho(z_i)_f$ to effectively penalize excessively high mutual information $I(f;z_i)$. The computation of $\rho(z_i)_f$ is detailed in Eq.16 of \cite{Hannun2021MeasuringDL}. In this context, suppressing $\rho(z_i)_f$ (approximately suppressing $I(f;z_i)$) is equivalent to penalizing the magnitude of the gradient of $f$ with respect to each $z_i$. This approach, supported by findings in \cite{smilkov2017smoothgrad}, uses gradient regularization to smooth the classifier's decision boundary, thereby reducing the potential risk of overfitting and enhancing adversarial resilience.
> >
> >
> > **Minimizing $\Psi(x_{i,\omega_i},x_{i,\overline{\omega_i}})$ and $\Phi(x_{i,\omega_i},\hat{x}_{i,{\omega_i}})$ via smoothing the distribution of feature-wise contribution to the classification output.** The primary goal of regularizing these terms is to prevent the classifier from relying too heavily on a few influential features. To achieve this, we propose using Integrated Gradient (IG) \cite{sundararajan2017axiomatic} to assess feature-wise contributions to the classification output. We apply Total-Variance (TV) regularization over the feature-wise Integrated Gradient to promote a smooth and balanced distribution of feature-wise attribution. In Appendix I.1, we show empirically with toy models that
> > performing the proposed TV regularization can reduce the estimated value of both mutual information-based terms.
> >
> >
> > [Hannun2021MeasuringDL] Hannun A, Guo C, van der Maaten L. Measuring data leakage in machine-learning models with Fisher information[C]//Uncertainty in Artificial Intelligence. PMLR, 2021: 760-770.
> >
> > [wei2016mutual] Wei X X, Stocker A A. Mutual information, Fisher information, and efficient coding[J]. Neural computation, 2016, 28(2): 305-326.
> >
> > [smilkov2017smoothgrad] Smilkov D, Thorat N, Kim B, et al. Smoothgrad: removing noise by adding noise[J]. arXiv preprint arXiv:1706.03825, 2017.
> >
> > [sundararajan2017axiomatic] Sundararajan M, Taly A, Yan Q. Axiomatic attribution for deep networks[C]//International conference on machine learning. PMLR, 2017: 3319-3328.

---

> ### Author Response · Authors · 2023-11-19
> **Evaluating the upper bound with practical data and models**
>
> **Evaluating the upper bound with practical data and models**: We emphasize that evaluating $I(f;z_{i})$ suffers from computational complexity and potential inaccuracy due to the curse of dimensionality. According to \cite{Gao2018TIT}, it is impossible to construct a consistent mutual information estimation for high dimensional random variables, e.g. the parameters of deep neural networks with complex architectures.
>
> We can only approximate this mutual information term for simple model architectures, such as logistic regression. We thus follow the approximated computation method to the mutual information-based bounds proposed in \cite{Bu et.al., 2020}. In this setting, we focus on a simplified model comprising a single fully connected layer with an activation function. As for the activation function, we use the softmax function for multi-class classification and the sigmoid function for binary classification. In the revised Appendix I.1 of our latest manuscript, we provide the approximated evaluation of the mutual information bound in Eq.3.
>
> The methodology comprises $M$ iterations of model training. In each iteration, $N$ samples are randomly selected from the training set and trained to convergence using IGSG and Std Train techniques. Following training, samples and model weights from each iteration are aggregated. Subsequently, the Mutual Information Neural Estimation (MINE) technique, as referenced from Belghazi et al. (2018), is applied to evaluate the terms and their weighted sum in Eq.3. A critical aspect of this calculation is the term $\omega_i$, denoting the perturbed features. Due to the impracticality of identifying the most vulnerable features for each sample and its corresponding classifier, $\omega_i$ is predetermined. This is done by attacking the MLP model trained with Std Train and selecting the top 5 features based on the highest attack frequency as $\omega_i$, which is then fixed for the entire dataset.
>
> The mutual information bounds for Splice and PEDec, trained using IGSG and Std Train (vanilla gradient descent without performing the IGSG-based regularization), are analyzed. The results are depicted in Figure 6. In summary, we observe that the classifier trained with IGSG exhibits lower values for all four mutual information terms in the proposed upper bound in Eq.3 (thus a globally lower bound value) and higher adversarial accuracy across the two datasets. This finding firstly indicates that enforcing IGSG regularization can reduce the mutual information-based upper bound of the adversarial risk proposed in Eq.3. Furthermore, we consider adversarial accuracy as a measure of actual adversarial risk. Higher adversarial accuracy indicates lower adversarial risk and vice versa. This quantitative evaluation demonstrates the correlation between the upper bound and actual adversarial risk. Lower values of the mutual information bound signify higher adversarial accuracy, thus indicating a reduced level of adversarial risk.
>
> [Gao2018TIT] W.Gao,S.Oh,and P.Viswanath,“Demystifying fixed k-nearest neighbor information estimators,”IEEE Trans.Inform.Theory,vol.64, no.8,pp.5629–5661,2018.
>
> [Bu et.al., 2020] Bu Y, Zou S, Veeravalli V V. Tightening mutual information-based bounds on generalization error[J]. IEEE Journal on Selected Areas in Information Theory, 2020, 1(1): 121-130.
>
> [Belghazi et al. 2018] Belghazi M I, Baratin A, Rajeshwar S, et al. Mutual information neural estimation[C]//International conference on machine learning. PMLR, 2018: 531-540.

---

> ### Author Response · Authors · 2023-11-19
> **Tightness of the upper bound**
>
> **Tightness of the upper bound**: We provide the discussion about the tightness of the proposed adversarial risk upper bound at the end of Appendix A, as highlighted by the texts with blue fonts.
>
> First, we show this bound reduces to a tight individual sample mutual information based upper bound of the generalization error of $f$ in the adversary-free case. It converges to zero when $n\rightarrow\infty$ with the same speed as the generalization bound established in Proposition.1 of \cite{Bu et.al., 2020}. This bound enjoys a close level of tightness in the adversary-free scenario as that proposed in \cite{Bu et.al., 2020}. It is tighter than the mutual information-based generalization bound developed in \cite{Xu2017NiPS}.
>
> Second, we show that the value of Eq.3 is always bounded from above, as shown in Eq.16 at the end of Appendix A. In Eq.16, the maximum cardinality of any single feature in the feature subset $\omega_i$ is denoted as q. $\epsilon$ is the maximum number of features that the attacker may perturb, a.k.a the attack budget. the number of the features in $\omega_i$, noted as $|\omega_i|$ is no more than $\epsilon$. On the right side of this inequality, the first term under the square root symbol is $\sum_{i=1}^n3I(f;z_i)$. It measures the generalization error under the adversary-free setting. The second term $\log(q\epsilon)$ measures the strength of the attack by considering the cardinality of the feature subset $\mathcal\omega_i$. A higher cardinality $\log(q\epsilon)$ implies a larger combinatorial set of possible categorical feature values available to the attacker (more features that the attacker may perturb and/or more category values per feature that the attacker may choose to replace the original feature value). The attacker selects one set of categorical values in this combinatorial set to replace the original feature values within the feature subset $\omega_i$, in order to deliver the adversarial attack. Consequently, a higher cardinality indicates greater flexibility to organize feature manipulation over $\omega_{i}$, which signifies a stronger attack and thereby elevates the adversarial risk. The right side of this inequality gives a bounded but rough estimate of the adversarial risk, as not all of the features are useful for attack. Only the perturbation over influential features may effectively cause the rise of adversarial risk. In this sense, our proposed bound provides a more accurate estimate of the actual adversarial risk.
>
> [Bu et.al., 2020] Bu Y, Zou S, Veeravalli V V. Tightening mutual information-based bounds on generalization error[J]. IEEE Journal on Selected Areas in Information Theory, 2020, 1(1): 121-130.
>
> [Xu2017NiPS] Aolin Xu and Maxim Raginsky. 2017. Information-theoretic analysis of generalization capability of learning algorithms. In Proceedings of the 31st International Conference on Neural Information Processing Systems (NIPS'17). Curran Associates Inc., Red Hook, NY, USA, 2521–2530.

---

### Official Review · Reviewer_oxAy · 2023-11-01

**Soundness:** 2 fair
**Presentation:** 1 poor
**Contribution:** 2 fair
**Rating:** 3
**Confidence:** 4

**Summary:**

This paper studies how to strength the adversarial robustness of categorical data using attribution smoothing.

**Strengths:**

- There have been only a few works aiming to enhance the adversarial robustness of categorical data.

- The empirical results look quite convincing.

- The use of IG for discrete data is interesting.

**Weaknesses:**

- The writing needs a significant boost and improvement.  The notions used in this paper are not good and easy to confuse such $b(x)_{i,j,k}$ sometimes $x_{i,j,k}$ for the element $jk$ of the categorical data point $x_i$.

- Theorem 1 has unsolid terms without a careful explanation
   -  $I(f;S^n)$ with the classifier $f$ and the training set $S^n$, how can you evaluate the mutual information between a classifier and a training set?
   -  $I(x_{i,\omega_i}; f)$  with $\omega_i$ to be the selected feature, however, it is not clear  $x_{i,\omega_i}$ and how to compute $I(x_{i,\omega_i}; f)$ because it seems that $f(x_{i,\omega_i})$ is not valid. Similar doubt for $I(x_{\bar{\omega}_i, y_i; f})$.

-  The theories developed and the proposed approach are not really connected.

**Questions:**

I believe this paper has some interesting ideas. However, the presentation and writing need a significant boost, hence it is not ready to publish.

For questions, please refer to the weaknesses and
- Can you further explain more the intuition of the term $\ell_{TV}IG(x_i)$?

---

> ### Author Response · Authors · 2023-11-19
> **Change of Notations**
>
> Many thanks to the viewer for the insightful questions. In the response below, we first provide answers to these questions, and then would like to discuss more about the changes we made in the updated draft, thanks to the insightful questions from the reviewer.
>
> **Notations**: Thank you for the advice about the notation. We have updated the usage of superscript and subscript for indexing the feature and categorical values in each sample, so the representation is more consistent and understandable. In this updated notation, $x_i$ represents a sample, and $b(x_i)$ signifies the one-hot encoding of $x_i$.
>
> In $b(x_i)_{j,k}$, the subscript $j$ refers to feature $j$, and $k$ indicates categorical value $k$ taken by feature $j$.
>
> For example, if the $j$-th feature of sample $x_i$ takes the $k^*$-th value, $b(x_i)_{j, k^*}=1$,
>
> and for all other $k\neq k^*$, $b(x_i)_{j,k}=0$. We hope that this updated notation can help readers better comprehend the relationships and operations being described in our paper.

---

> ### Author Response · Authors · 2023-11-19
> **The mutual information evaluation of $I(f;S^n)$**
>
> **The mutual information evaluation of $I(f;S^n)$**: $I(f; S^n)$  in  Theorem 1  is an information-theoretic bound for generalization error. Akin to VC dimension, the quantitative evaluation of $I(f; S^n)$ proves challenging in general scenarios, as discussed in \cite{Xu2017NiPS}. The reviewer asked a great question. Computing $I(f;S^n)$ with the classifier $f$ and the training set $S^n$ needs to calculate the mutual information between two high-dimensional random vectors. Due to the curse of dimensionality, it is impossible to construct a consistent mutual information estimation for a large $n$ \cite{Gao2018TIT}.
>
> To make $I(f; S^n)$ be computationally feasible, one possible approximation is to calculate it  based on $I(f;z_{i})$, which is the mutual information calculated for individual samples $z_{i}=\{(x_i,y_i)\}$. We have updated Theorem 1 and analyzed the adversarial risk upper bound based on this approximation. Please refer to our revision in Section 3 highlighted with the blue color font and corresponding proof in Appendix A. The details about the approximate computation can be found in Appendix I.1. The updated adversarial risk upper bound with the individual sample mutual information $I(f;z_{i})$ leads to the same analysis about the main factors of suppressing adversarial risk on categorical data and the rationality of the proposed IGSG regularization technique.
>
>
> [Xu2017NiPS] Aolin Xu and Maxim Raginsky. 2017. Information-theoretic analysis of generalization capability of learning algorithms. In Proceedings of the 31st International Conference on Neural Information Processing Systems (NIPS'17). Curran Associates Inc., Red Hook, NY, USA, 2521–2530.
>
> [Gao2018TIT] W.Gao,S.Oh,and P.Viswanath,“Demystifying fixed k-nearest neighbor information estimators,”IEEE Trans.Inform.Theory,vol.64, no.8,pp.5629–5661,2018.

---

> ### Author Response · Authors · 2023-11-19
> **Explanations of $x_{i,\omega_i}$ and   $I(x_{i,\omega_i};f)$**
>
> **Explantions of $x_{i,\omega_i}$**: As given in Theorem 1, for each sample $x_{i}$, the set of the categorical features modified by the adversary and the rest untouched features are noted as $\omega_{i}$ and $\overline{\omega_{i}}$, respectively.
> $x_{i,\omega_i}$, therefore, denotes the features that the adversary modifies to deliver the attack on the data point $x_{i}$. For example, on Splice dataset, each sample $x_i$ contains $60$ features. One possible combination of perturbed features may be $\\{1,29,31\\}$, which means that only these three features are perturbed and all other features remain unchanged. In this case $\omega_i = \\{1,29,31\\}$ and $\overline{\omega_i}=\\{2,3,...,28,30,32,33,...,60\\}$.
>
> **Explantions of $I(x_{i,\omega_i};f)$**: $I(x_{i,\omega_i};f)$ denotes the mutual information between the feature subset $\omega_{i}$ of the data point $x_i$ and the hypothesis $f$. The computation of $I(x_{i,\omega_i};f)$ is carried out in a similar manner to $I(x_i;f)$, with the key difference being that it utilizes only the features within $\omega_{i}$, as opposed to employing all available features. Similarly, $I(x_{i,\overline{\omega_i}};f)$ is computed using only the features that are not part of $\omega_{i}$.

---

> ### Author Response · Authors · 2023-11-19
> **Connections between the developed theories and the proposed approach**
>
> **Connections between the developed theories and the proposed approach**: The developed theories in Section 3 reveal two important factors to suppress the adversarial risk over categorical inputs. The proposed approach in Section 4 is in accordance with two recommended factors to minimize the adversarial risk. Taking into account Factor 1, we  penalize $I(z_i;f)$ by enforcing the regularization over Fisher information of the classifier $f$ for the data point $z_i$, so the adversarial risk is reduced. Inspired by Factor 2, our goal is to smooth out the distribution of feature-wise contributions to the classification result, thereby decreasing the model's reliance on a small number of influential features. We thus implement the Integrated Gradient technique \cite{SundararajanIG2017} to evaluate the feature-wise contributions. Additionally, we adopt the Total Variation (TV) loss, as outlined by \cite{ChambolleJMIV2004}, over the feature-wise Integrated Gradient. To further clarify these connections, we updated the paragraphs in Section 3, Section 4 and Appendix B (highlighted in blue).
>
>
> [SundararajanIG2017] Mukund Sundararajan, Ankur Taly, and Qiqi Yan. 2017. Axiomatic attribution for deep networks. In Proceedings of the 34th International Conference on Machine Learning - Volume 70 (ICML'17). JMLR.org, 3319–3328.
>
> [ChambolleJMIV2004] Antonin Chambolle, An Algorithm for Total Variation Minimization and Applications. Journal of Mathematical Imaging and Vision 20, 89–97 (2004). https://doi.org/10.1023/B:JMIV.0000011325.36760.1e

---

> ### Author Response · Authors · 2023-11-19
> **The intuition of $\ell_{TV}IG(x_i)$**
>
> **The intuition of $\ell_{TV}IG(x_i)$**: The design of Total Variation (TV) loss regularization towards Integrated Gradient (IG) scores of each feature is to reduce the total variation of the IG score of each feature and smooth away spurious noise with significantly higher values compared to the normal value range of the IG score. This TV loss regularization of the IG scores corresponds to the leverage of Factor 2 in Section 3.2. Applying this TV regularization is to ensure a more uniform distribution of these contributions in the classifier’s decision process. In Appendix I.1, we show empirically with the toy model that performing the proposed TV regularization indeed reaches the expected goal: reducing both mutual information-based terms, $\Psi(x_{i,\omega_i}, x_{i,\overline{\omega_i}})$ and $\Phi(x_{i,\omega_i}, \hat x_{i_{\omega_i}})$, in the adversarial risk bound to smooth the distribution of feature-wise contribution to the classification output. Furthermore, a comparison of the TV loss and the $L_2$ loss is presented in Appendix I.6, demonstrating that the TV loss guided IGSG consistently surpasses the $L_2$ loss guided $L_2$-IGSG in performance.

---

> ### Author Response · Authors · 2023-11-19
> **Approximated computation of the mutual information terms**
>
> The thoughtful questions from the reviewer inspired us to provide an in-depth analysis of the evaluation of mutual information $I(f; S^n)$ and its approximation $I(f;z_{i})$. We have included these discussions in Section 3.2 and the proof in Appendix A, and would like to share them here with Reviewer oxAy if any further discussion is useful for addressing the concerns that Reviewer oxAy has.
>
> **Approximated computation of the mutual information terms**: Appendix I.1 on Page 25  provides the approximated computation of the upper bound in Eq.3. It is worth noting that the approximate evaluation $I(f;z_{i})$ for $I(f; S^n)$  still suffers from computational complexity and potential inaccuracy due to the curse of dimensionality. We can only approximate $I(f; S^n)$  for simple model architectures, such as logistic regression. To showcase the approximate evaluation of a toy model, we take a  simplified model comprising a single fully connected layer, where the output activation function is the softmax function for multi-class classification and the sigmoid function for binary classification. In each iteration of training, $N$ samples are randomly selected from the training set and used to train the toy model until convergence using either standard training or our proposed IGSG  techniques. Based on the samples and model weights from each iteration, the Mutual Information Neural Estimation (MINE) technique, as referenced from \cite{Belghazi et al. 2018}, is employed to evaluate the terms and their weighted sum in Eq.3. Regarding  $\omega_i$,  the most vulnerable features for each sample and its corresponding classifier, they are obtained by attacking the standard trained toy model and selecting the top 5 features based on the highest attack frequency. The results of $I(f;z_{i})$ evaluated for this toy model trained on Splice and PEDec dataset are shown in Figure 6 of Appendix I.1 on Page 25.
>
> **$I(f;z_{i})$: standard training vs our IGSG training**: We observe in Figure 6 that, compared to the standard training, the model trained with IGSG exhibits lower values for all the four mutual information terms in the proposed upper bound in Eq.3 (thus a globally lower bound value) and higher adversarial accuracy across the two datasets. This finding indicates that enforcing IGSG regularization is effective in reducing the mutual information-based upper bound of the adversarial risk proposed in Eq.3.
>
>
> [Belghazi et al. 2018] Belghazi M I, Baratin A, Rajeshwar S, et al. Mutual information neural estimation[C]//International conference on machine learning. PMLR, 2018: 531-540.

---

> ### Author Response · Authors · 2023-11-19
> **The benefits of shaping the adversarial risk upper bound with $I(f;z_i)$**
>
> **The benefits of shaping the adversarial risk upper bound with $I(f;z_i)$**: The benefits are twofold. First, compared to the training set mutual information $I(f; S^n)$, the bound based on individual sample mutual information in Eq.3 poses a tighter bound over the generalization error, as evidenced by the analysis in \cite{Bu et.al., 2020}. We provide the discussion about the tightness of the bound using $I(f;z_i)$ in Appendix A. The tighter bound results largely from discarding the previous assumption made for $I(f; S^n)$, which is that the loss function $\ell(f,z)$ has a bounded cumulative generating function with $z\sim\mu_{z}$ and $f\in{\mathcal{H}}$ \cite{Xu2017NiPS}. However, this assumption does not necessarily hold in practice.
> Second, according to \cite{Bu et.al., 2020}, $I(f;S^n)$ may have infinite values. In contrast, $I(f;z_{i})$ is bounded and can be estimated quantitatively under certain constraints to the model complexity of $f$.  Concretely, we use a toy model, i.e. a logistic regression model, to conduct empirical estimation to the adversarial risk bound proposed in Eq.3 on Splice and PEDec data. The details about empirical evaluation are provided in Appendix I.1.
>
> We compute the estimation of the risk bound of this toy model trained with and without using the proposed IGSG robust training method. The results in Appendix I.1 can be summarized from two perspectives. 1) We find that with IGSG deployed, all of the three components in the adversarial risk bound decrease on Splice and PEDec data. This result confirms that enforcing IGSG-based regularization reaches the expected goal: IGSG minimizes the adversarial risk bound by minimizing both $\sum_{i=1}^n I(f;z_i)$ and $\sum_{i=1}^{n} 2\Psi + \Phi$.  2) We observe that a lower value of the adversarial risk bound consistently corresponds to a higher adversarial accuracy on both datasets. This indicates that minimizing the adversarial risk bound in Eq.3 can suppress the actual adversarial risk, thus improving the classification accuracy over adversarially modified samples.
>
> [Bu et.al., 2020] Bu Y, Zou S, Veeravalli V V. Tightening mutual information-based bounds on generalization error[J]. IEEE Journal on Selected Areas in Information Theory, 2020, 1(1): 121-130.
>
> [Xu2017NiPS] Aolin Xu and Maxim Raginsky. 2017. Information-theoretic analysis of generalization capability of learning algorithms. In Proceedings of the 31st International Conference on Neural Information Processing Systems (NIPS'17). Curran Associates Inc., Red Hook, NY, USA, 2521–2530.

---

### Author Response · Authors · 2023-11-19
**Summary of the changes**

We deeply appreciate all the constructive comments and suggestions from 6 reviewers. We have updated our submission to address the raised questions. The updated / added contents are highlighted in blue. Globally, our revision is summarized below:

We have updated Section 3.2, 4 and Appendix B (Page.16 of the revised submission) to clarify the connection between the information-theoretic upper bound and the design of IGSG. We also discussed the tightness of the derived upper bound at the end of Appendix A (Page 15-16 of the revised submission).

In Appendix C and D (Page 17 of the revised submission), we discuss the difference between our study and PAC-Bayes bounds, and the rationality behind the assumption of a randomized learning mechanism.

In Appendix I.1 (Page 25-26 of the revised submission), we have added approximated computation to the upper bound proposed in Eq.3 to confirm empirically the relation between minimizing the bound and improving the adversarial robustness of the target classifier.

We have compared the visualization of the classification boundary before and after performing the proposed IGSG-based robust training. We show that IGSG indeed smooths the classification boundary. The results are presented in Appendix I.2 (Page 26-27 of the revised submission).

In Appendix I.7 (Page 30 of the revised submission), we apply PCAA attack on IGSG and all the baseline methods across all three datasets on two models. The results show that IGSG is still effective under PCAA attack. Also, we augment PCAA-based adversarial training (PAdvT) as another baseline model. We compare its performance with IGSG’s performance in Section 5.2 and Appendix I.3 (Page 27 of the revised submission). IGSG consistently has superior performance than PAdvT across all the datasets.

Our detailed response to the raised questions from each of the reviewers can be found below the corresponding comments. We will also adjust the paper to enhance readability.

---

> ### Author Response · Authors · 2023-11-23
> **Additional Changes**
>
> In addition to the aforementioned updates, we have also evaluated the sensitivity of the hyperparameters on the test set. The results of this analysis are detailed in our response to reviewer iddk.

---

### Author Response · Authors · 2023-11-21
**Any suggestion regarding our revision**

Dear reviewers, we deeply appreciate all of your helpful review comments. We hope our response and revision can address the questions you raised. But please don't hesitate to let us know, if you have further comments regarding our answers. We are more than happy to dive into discussions.

---

### Meta-Review · Area_Chair_ELRt · 2023-12-05

**Metareview:**

This work proposes a regularization approach to adversarial robustness with respect to categorical features.  Reviewers noted that the theoretical contributions are not closely related to the proposed methodology and are unnecessary for developing or understanding the straightforward method.  The reviewers also noted that the presentation could use improvement.  This work also conducts experiments on few datasets and models and makes moderate improvements on a problem that is not of broad interest to the community.  Therefore, I am inclined to reject this paper, but I believe that with improved presentation and experiments, this could make a strong submission.

**Justification For Why Not Higher Score:**

There is nothing wrong per se with this paper, but it contains irrelevant theory work, limited evaluations, and is does not approach a problem that has broad interest.

**Justification For Why Not Lower Score:**

N/A

---

### Decision · Program_Chairs · 2024-01-16

Reject